# Demystifying Entropy Control in LLM RL Training: Theoretical Analysis and Dynamic Scheduling

Jingchu Gai [*1]  Guanning Zeng [*2]  Huaqing Zhang [2]  Han Zhong [34]  Yige Hong [1]  Andrej Risteski [1]
Aditi Raghunathan [1]

## Abstract

We investigate a pivotal yet debated component of reinforcement learning (RL) for training large language models (LLMs): controlling entropy (increasing or decreasing it) during RL fine-tuning. The existing literature presents a dichotomy: some studies posit that increasing entropy facilitates exploration, whereas others argue that decreasing entropy enhances performance. Crucially, we observe that the impact of entropy regularization exhibits significant heterogeneity across different tasks. In this paper, we resolve this conflict by identifying the governing factor of optimal entropy control. We define *Entropy Discrepancy* (Definition 1) and demonstrate that this metric dictates the appropriate direction of regularization. Guided by this insight, we derive a principled dynamic scheduling method that adaptively modulates the entropy coefficient, seamlessly switching between maximization and minimization as training evolves. Extensive experiments confirm the correlation between Entropy Discrepancy and the efficacy of entropy control. Furthermore, our adaptive method yields substantial improvements over vanilla RL, achieving a 6.7 percentage-point Pass@K gain on AIME24 and a 17.52 percentage-point gain on KNK5 at Pass@1.

## 1. Introduction

Controlling policy entropy plays a pivotal role in reinforcement learning (RL) (Schulman et al., 2017; Haarnoja et al., 2018). As RL fine-tuning has emerged as an effective paradigm for improving Large Language Models (LLMs), controlling entropy in this context has become similarly critical (Cui et al., 2025b; Liu et al., 2025; Wang et al., 2025). In traditional RL, entropy maximization is a standard strategy to encourage exploration, preventing the agent from becoming trapped in local minima (Mnih et al., 2016; Schulman et al., 2017). From a theoretical perspective, this practice is grounded in standard learning theory regularization principles, which demonstrate that entropy regularization yields sharper statistical convergence rates (Geist et al., 2019; Mei et al., 2020; Cen et al., 2022). Analogously, this principle extends to LLM training. Numerous recent works advocate for actively maintaining high entropy levels during fine-tuning. These approaches aim to prevent entropy collapse, a phenomenon that precipitates a rapid loss of exploration capability, severely impairing model performance and diminishing generation diversity.

However, some recent studies also observe the opposite trend: decreasing entropy can actually enhance the reasoning performance of LLMs (Agarwal et al., 2025; Gai et al., 2026; Gao et al., 2025). This phenomenon challenges traditional RL theory—which typically advocates for entropy maximization and the theoretical underpinnings of the benefits of entropy minimization remain largely unexplored.

To better understand this, we first empirically compare entropy minimization vs entropy maximization starting with the same model and using the same dataset on several tasks. We observe that the optimal entropy strategy is highly task-dependent (see Figure 1). In some tasks, increasing entropy boosts both Pass@1 and Pass@K metrics; in others, decreasing entropy yields better performance. Motivated by these disparate observations, we study the following questions: (1) **Mechanism:** What factors drive these task-dependent variations, and specifically, what structural properties of the data govern this mechanism? (2) **Algorithmic Principles:** Based on the mechanisms we identify, is there a principled adaptive schedule for both the sign and scale of entropy

---
[*]Equal contribution  [1]Carnegie Mellon University, Pittsburgh, PA, USA. {jgai,yigeh,aristesk}@andrew.cmu.edu, raditi@cmu.edu  [2]Tsinghua University, Beijing, China. {zhanghq22,zgn21}@mails.tsinghua.edu.cn  [3]Antai College of Economics and Management, Shanghai Jiao Tong University, Shanghai, China. hanzhong.work@gmail.com  [4]Center for Data Science, Peking University, Beijing, China. Correspondence to: Jingchu Gai <jgai@andrew.cmu.edu>.

*Proceedings of the 43rd International Conference on Machine Learning*, Seoul, South Korea. PMLR 306, 2026. Copyright 2026 by the author(s).

*Figure 1. Illustration of how the effect of entropy varies in different cases*

regularization throughout training?

We address these open questions by establishing a simplified yet empirically predictive theoretical setting that identifies the sufficient conditions that characterize when entropy maximization and when entropy minimization enhances LLM performance during RL fine-tuning. Guided by these theoretical insights, we derive a method that dynamically adjusts the entropy regularization coefficient throughout the training process. We provide a detailed exposition of our contributions as follows.

**Governing factor of entropy control.** We introduce Entropy Discrepancy, defined as the difference between the base model's average entropy on positive versus negative samples. Our theoretical analysis (Theorem 2) establishes that increasing entropy is beneficial when Entropy Discrepancy is high, whereas decreasing entropy is preferred when this discrepancy is low. These findings further identify two distinct regimes where entropy minimization is effective: (1) when the base model exhibits high concentration (confidence) on correct samples, and (2) when the model demonstrates low diversity across both positive and negative samples.

**Dynamic Entropy Scheduling.** Building on this insight, we propose an adaptive scheduling algorithm that dynamically modulates the entropy coefficient throughout training. Specifically, we utilize the computed *Entropy Discrepancy* to govern the regularization term, thereby increasing entropy when the discrepancy is high and suppressing it when low. As the model's distribution over positive and negative samples evolves, our method automatically recalibrates the regularization strength in real-time to sustain optimality. Crucially, we provide a theoretical guarantee that this dynamic approach strictly outperforms static vanilla RL algorithms (Theorem 3).

**Empirical Validation.** We validate our framework using the Countdown task (Pan et al., 2025). By adjusting task specifications, we systematically modulate the intrinsic difficulty and solution density, thereby simulating scenarios with varying entropy demands. Using this setup, we examine the correlations between Entropy Discrepancy and the performance impact of increasing entropy, as well as that of decreasing entropy. Across all evaluated settings, we

achieve an absolute Spearman coefficient exceeding 0.8, empirically validating our theoretical postulate. Subsequently, we evaluate our adaptive entropy schedule on two distinct reasoning domains using three model architectures. Despite the inherent heterogeneity in how entropy affects different tasks and models, our adaptive method broadly yields superior performance compared to baseline approaches. Our dynamic approach achieves robust gains over vanilla RL, yielding a 17.52 percentage-point gain on KNK5 at Pass@1 and a 6.7 percentage-point gain on the AIME24 mathematical reasoning benchmark. Crucially, our method matches or surpasses the best fixed entropy coefficient identified via extensive grid search, highlighting the practical advantage of adaptive scheduling over static regularization. Furthermore, we benchmark our approach against recent methods designed to enhance RL reasoning. Remarkably, our method matches or even surpasses the performance of these complex baselines while relying solely on a global entropy schedule, offering a significantly more streamlined solution.

*Furthermore, our method surpasses both fixed entropy baselines and recent state-of-the-art reasoning methods.*

---

**Summary of Our Main Contribution**

We summarize our main contributions as follows:

1. **Theoretical Explanation for Negative Entropy.** We derive a rigorous theory in a simplified, yet empirically predictive setting explaining the counter-intuitive effectiveness of negative entropy in LLM fine-tuning, offering a unified perspective on various entropy control strategies.

2. **Factors affecting the effect of Entropy.** We identify *Entropy Discrepancy* as the governing factor for entropy's impact, deriving a principled criterion for entropy control.

3. **Dynamic Entropy Scheduling.** Guided by our theory, we propose a dynamic entropy scheduling algorithm. Empirical results on reasoning benchmarks demonstrate that our approach achieves robust gains over vanilla RL and remains competitive with or better than fixed entropy coefficient baselines.

**Conflict of Interest Disclosure.** The authors are affiliated solely with academic institutions (Carnegie Mellon University, Tsinghua University, and Peking University) and report no financial conflicts of interest related to this work. All experiments are conducted on publicly released, open-weight models (Qwen2.5, Qwen2.5-Math) and open datasets; the paper does not evaluate any proprietary model developed by an entity employing any of the authors.

## 2. Related Work

Reinforcement fine-tuning for LLMs is a nuanced process that involves several widely studied ingredients, including surrogate objectives (Tang et al., 2025; Walder & Karkhanis, 2025), policy-gradient baselines (Ahmadian et al., 2024; Li et al., 2024), reward engineering (Cui et al., 2025a; Bhaskar et al., 2025; Shao et al., 2025), entropy mechanisms (Agarwal et al., 2025; Gao et al., 2025; Zhang et al., 2025) and so on. Among these components, entropy regularization has long been a cornerstone of reinforcement learning, primarily employed to encourage exploration and smooth non-convex optimization landscapes (Haarnoja et al., 2018; Ahmed et al., 2019; Mei et al., 2020). In the context of Large Language Models (LLMs), alignment algorithms like PPO (Schulman et al., 2017) and GRPO (Shao et al., 2024) typically enforce trust regions via KL-divergence or explicit bonuses to maintain generation diversity and prevent premature mode collapse (Ouyang et al., 2022). However, recent empirical findings in complex reasoning tasks challenge this maximization paradigm. Several studies report the counter-intuitive phenomenon that suppressing entropy (i.e., entropy minimization) can enhance reasoning accuracy (Agarwal et al., 2025; Gao et al., 2025), while others suggest that dynamic schedules—alternating between increasing and decreasing phases—outperform monotonic adjustments (Zhang et al., 2025; Gai et al., 2026). Despite these observations, a theoretical framework explaining *why* entropy minimization aids reasoning and *when* to switch scheduling phases remains absent. We comprehensively review the entropy mechanisms and their theoretical foundations in Appendix A.

## 3. Preliminary

In this section, we present the theoretical background and establish the notation used throughout the paper. We emphasize that the policy parameterization and training dynamics described herein serve as a tractable abstraction for our theoretical analysis, rather than a direct description of the practical implementation. We employ this simplified framework to mathematically characterize the effects of entropy and to derive principled algorithmic interventions.

**Notations and Basic Setup.** Let $x \in \mathcal{X}$ denote a given input prompt. Let $\mathbf{y} = (y_1, y_2, \ldots, y_T) \in \mathcal{Y}$ represent the corresponding response. The language model is parameterized as a policy $\pi$, from which responses are sampled: $\mathbf{y} \sim \pi(\cdot \mid x)$. We refer to the prompt-response pair $(x, \mathbf{y})$ as a *trajectory* and denote it as $\tau$. Each trajectory receives a terminal reward $r(\tau)$. Let $\mathcal{C} \subseteq \mathcal{Y}$ denote the set of ground-truth correct responses. We adopt a binary reward function defined as $r(\tau) = \mathbf{1}[\tau \in \mathcal{C}]$. The objective of reinforcement learning is to maximize the expected reward, defined as $V(\pi) = \mathbb{E}_{\tau \sim \pi}[r(\tau)]$. For any distribution $p$ over the support set $\mathcal{S} \subseteq \mathcal{Y}$, we denote its entropy by $\mathcal{H}(p)$, defined as: $\mathcal{H}(p) = -\sum_{\tau \in \mathcal{S}} p(\tau) \ln p(\tau)$.

**Policy Parameterization.** We adopt the standard softmax parameterization (also known as the Gibbs policy) to model the agent's behavior. The policy is a conditional distribution over responses given the prompt $x$, given by:

$$\pi_\theta(\mathbf{y} \mid x) = \frac{\exp(\theta(x, \mathbf{y}))}{\sum_{\mathbf{y}' \in \mathcal{Y}} \exp(\theta(x, \mathbf{y}'))},$$

where $\theta(x, \mathbf{y}) \in \mathbb{R}$ represents the unnormalized log-probability (or logit) of the trajectory $\tau = (x, \mathbf{y})$, parameterized by $\theta$. For brevity, we write $\pi_\theta(\tau)$ for $\pi_\theta(\mathbf{y} \mid x)$. This formulation is canonical in theoretical reinforcement learning analysis due to its favorable optimization properties, as it guarantees global convergence rates (Sutton & Barto, 1998; Mei et al., 2020; Cen et al., 2022). Furthermore, it serves as a tractable abstraction of the auto-regressive generation process in modern LLMs, which fundamentally operate via softmax distributions (Ziegler et al., 2019; Rafailov et al., 2023; 2024; Zhong et al., 2025).

**Training Dynamics.** The theoretical analysis that allows us to derive an entropy scheduling algorithm relies on analyzing Natural Policy Gradient (NPG) (Kakade, 2001). Let $\pi^{(t)}$, parameterized by $\theta^{(t)}$, denote the policy at the $t$-th iteration. The parameters $\theta$ of vanilla RL are updated with:

$$\theta^{(t+1)} = \theta^{(t)} + \eta \cdot \mathcal{F}_{\theta^{(t)}}^\dagger \nabla_\theta V(\pi^{(t)}). \tag{1}$$

Here, $\eta$ represents the learning rate, $(\cdot)^\dagger$ denotes the Moore-Penrose pseudoinverse, and $\mathcal{F}_\theta$ is the Fisher Information Matrix, defined as:

$$\mathcal{F}_\theta = \mathbb{E}_{\tau \sim \pi_\theta}[\nabla_\theta \ln \pi_\theta(\tau) \left(\nabla_\theta \ln \pi_\theta(\tau)\right)^\top].$$

NPG is used to effectively precondition the updates using the Fisher geometry, resulting in global convergence and potential acceleration (Agarwal et al., 2021; Shani et al., 2020; Cen et al., 2022). Our focus on NPG is motivated not only by its favorable convergence theory under softmax parameterization, but also by its connection to modern RL algorithms. In particular, in the softmax policy class, a natural policy gradient step can be equivalently viewed as solving a KL-regularized policy improvement problem or a mirror descent step on the probability simplex. This viewpoint (see e.g., Page 11 in Cai et al., 2020) unifies NPG with widely used PPO (Schulman et al., 2017) and TRPO

(Schulman et al., 2015) algorithms, and shows that all these algorithms admit the same policy update rule under the softmax parametrization: $\pi^{(t+1)}(\tau) \propto \pi^{(t)}(\tau) \exp(\eta r(\tau))$.

**Entropy-Regularized Updates.** For RL with positive and negative entropy regularization, the update rules are:

$$\text{Positive Entropy:} \quad \theta^{(t+1)} = \theta^{(t)} + \eta \cdot \mathcal{F}_{\theta^{(t)}}^\dagger \nabla_\theta V_+(\pi^{(t)}),$$
$$\text{Negative Entropy:} \quad \theta^{(t+1)} = \theta^{(t)} + \eta \cdot \mathcal{F}_{\theta^{(t)}}^\dagger \nabla_\theta V_-(\pi^{(t)}),$$

The corresponding objective functions $V_+(\pi)$ and $V_-(\pi)$ are defined as:

$$V_+(\pi) = \mathbb{E}_{\tau \sim \pi}[r(\tau)] - \rho_+ \cdot \sum_\tau \pi(\tau) \ln \pi(\tau),$$
$$V_-(\pi) = \mathbb{E}_{\tau \sim \pi}[r(\tau)] + \rho_- \cdot \sum_\tau \pi(\tau) \ln \pi(\tau),$$

where $\rho_+, \rho_- > 0$ denote the regularization coefficients for the positive and negative entropy settings, respectively. For the $t$-th training iteration, we denote the policy parameters for the vanilla, negative-entropy, and positive-entropy settings as $\theta_{\text{van}}^{(t)}$, $\theta_-^{(t)}$, and $\theta_+^{(t)}$, respectively. The corresponding policies are denoted by $\pi_{\text{van}}^{(t)}$, $\pi_-^{(t)}$, and $\pi_+^{(t)}$. We assume that all three algorithms are initialized from a common base model, denoted as $\pi_{\text{base}}$. Consequently, the initial policies satisfy $\pi_{\text{van}}^{(0)} = \pi_-^{(0)} = \pi_+^{(0)} = \pi_{\text{base}}$.

# 4. Theoretical Analysis of Negative and Positive Entropy Regularization

In this section, we provide a theoretical analysis of negative entropy regularization to identify the key factors governing its effectiveness. We begin by formally introducing the concept of Entropy Discrepancy. We then establish a fundamental theorem characterizing the regimes in which entropy maximization versus minimization is optimal, demonstrating that this optimality depends critically on the entropy discrepancy.

---

**Entropy Discrepancy**

**Definition 1.** *For a given policy $\pi \in \Delta(\mathcal{Y})$, we define its **Entropy Discrepancy** as:*

$$\text{Dis}(\pi) = -\frac{\sum_{\tau \in \mathcal{C}} \pi(\tau) \ln \pi(\tau)}{\sum_{\tau \in \mathcal{C}} \pi(\tau)} + \frac{\sum_{\tau \notin \mathcal{C}} \pi(\tau) \ln \pi(\tau)}{\sum_{\tau \notin \mathcal{C}} \pi(\tau)}.$$

---

Equipped with this definition, we now present a theoretical analysis of entropy regularization, demonstrating that its effectiveness hinges on the Entropy Discrepancy of the base model.

---

**Theoretical Analysis of Entropy Regularization**

**Theorem 2.** *Consider the reward function defined in Section 3. For a fixed step $T > 0$ and learning rate*

---

$\eta > 0$, *if the following condition holds:*

$$\tfrac{1}{2}\eta(T - 1) - \text{Dis}(\pi_{\text{base}}) > 0, \tag{2}$$

*then there exists a regularization coefficient $\rho_- > 0$ such that negative entropy regularization outperforms vanilla RL, i.e., $V(\pi_-^{(T)}) > V(\pi_{\text{van}}^{(T)})$. Conversely, if the inequality in Equation (2) does not hold, there exists a coefficient $\rho_+ > 0$ such that positive entropy regularization yields a higher value, i.e., $V(\pi_+^{(T)}) > V(\pi_{\text{van}}^{(T)})$.*

---

We sketch the proof by viewing $V(\pi_-^{(T)})$ as a function of $\rho_-$. Computing the first-order derivative at the origin, we find that $\frac{\partial V}{\partial \rho_-}\big|_{\rho_-=0} > 0$ iff Condition Equation (2) is satisfied. This positive gradient ensures that for sufficiently small $\rho_-$, the regularized policy strictly dominates the vanilla baseline, which proves the theorem.

According to Theorem 2, for a fixed training step $T$, the efficacy of entropy regularization is determined by the Entropy Discrepancy of the base model, which can be rewritten as:

$$\text{Dis}(\pi_{\text{base}}) = \underbrace{\ln(1 - p_{\text{base}}^{\checkmark}) - \ln(p_{\text{base}}^{\checkmark})}_{\text{Concentration}} + \underbrace{\mathcal{H}(\pi_{\text{base}}^{\checkmark}) - \mathcal{H}(\pi_{\text{base}}^{\times})}_{\text{diversity}},$$

where $p_{\text{base}}^{\checkmark} = \pi_{\text{base}}(\mathcal{C})$ is the total mass on correct samples. We define the conditional distributions $\pi_{\text{base}}^{\checkmark}$ and $\pi_{\text{base}}^{\times}$ by renormalizing $\pi_{\text{base}}$ over correct and incorrect sets, respectively. This decomposition highlights the key drivers of entropy regularization:

---

**Factors affecting the effect of Entropy**

1. **Concentration on correct samples.** Entropy minimization proves beneficial primarily when the base model already places a high probability mass on correct samples. This insight elucidates why the efficacy of negative entropy was historically overlooked. In traditional RL, policies are typically trained from scratch (random initialization), and existing theoretical analyses have predominantly focused on this low-accuracy regime where exploration is paramount. In contrast, LLM fine-tuning begins with a pre-trained model that already exhibits strong performance. Consequently, negative entropy regularization emerges as a superior strategy in this high-competence regime.

2. **Diversity on correct and incorrect samples.** Positive entropy regularization is particularly effective when the base model exhibits high diversity on correct samples and low diversity on incorrect samples. Intuitively, if the base model already cap-

---

tures a diverse array of correct answers, increasing entropy can further facilitate the exploration of the solution space. In contrast, when the model is initially concentrated on a limited subset of correct answers, it is more beneficial to optimize (exploit) the high-probability trajectories rather than enforcing exploration.

## 5. Dynamic Scheduling

Based on the analysis in Section 4, the regularization strategy is governed by the Entropy Discrepancy ($\mathrm{Dis}(\pi_{\mathrm{base}})$). Specifically, a large discrepancy warrants amplified entropy maximization, whereas a small (or negative) discrepancy favors entropy minimization to consolidate the policy.

Guided by this insight, we propose an algorithm that dynamically adapts the entropy coefficient during training. We further theoretically demonstrate that this dynamic scheduling outperforms both vanilla RL and fixed entropy regularization. To achieve this, we scale the regularization strength proportionally to the Entropy Discrepancy $\mathrm{Dis}(\pi_{\mathrm{base}})$ in Definition 1, optimizing the following objective function:

$$V_{\mathrm{dy}}(\pi) = \mathbb{E}_{\tau \sim \pi}[r(\tau)] + \rho_{\mathrm{dy}} \, \mathrm{Dis}(\pi_{\mathrm{no\ grad}}) \, \mathcal{H}(\pi),$$

where $\pi_{\mathrm{no\ grad}}$ denotes a copy of the policy $\pi$ that is treated as a constant during the gradient update (i.e., applying the stop-gradient operator). This mechanism is readily implemented in modern auto-differentiation frameworks. The parameters $\theta_{\mathrm{dy}}^{(t)}$ are updated via the Natural Policy Gradient rule defined in Equation (1): $\theta_{\mathrm{dy}}^{(t+1)} = \theta_{\mathrm{dy}}^{(t)} + \eta \cdot \mathcal{F}_{\theta_{\mathrm{dy}}^{(t)}}^{\dagger} \nabla_{\theta} V(\pi_{\mathrm{dy}}^{(t)})$, where $\pi_{\mathrm{dy}}^{(t)}$ represents the policy at the $t$-th iteration of our dynamic scheduling framework. We now proceed to the theoretical analysis of this method.

### Theoretical Analysis of Dynamic Scheduling

**Theorem 3.** *Consider the reward function defined in Section 3. For any fixed time step $T > 0$ and learning rate $\eta > 0$, there exists a regularization coefficient $\rho_{\mathrm{dy}} > 0$ such that the policy trained via dynamic entropy scheduling achieves a strictly higher value than the standard RL baseline, i.e., $V(\pi_{\mathrm{dy}}^{(T)}) > V(\pi_{\mathrm{van}}^{(T)})$.*

Theorem 3 establishes that while the effectiveness of static entropy regularization is contingent upon specific Entropy Discrepancy conditions, dynamic scheduling circumvents this requirement to surpass vanilla RL. Consequently, dynamic scheduling demonstrates superior performance across a broader range of scenarios. In the subsequent sections, we empirically validate both the proposed algorithm and the

underlying principle of entropy control via Entropy Discrepancy (Definition 1).

## 6. Experimental Evaluation of Entropy Control Principle

In this section, we empirically validate our theoretical analysis and the proposed method. We commence with a brief overview of Group Relative Policy Optimization (GRPO) (Shao et al., 2024), which serves as the backbone for our experiments. Building on this framework, we instantiate the Entropy Discrepancy metric and implement the Dynamic Entropy Scheduling proposed in Section 5.

Our evaluation proceeds in two steps. First, we investigate the correlation between Entropy Discrepancy and the performance gains yielded by static entropy regularization. Through controlled experiments across various tasks, we demonstrate a strong correlation (evidenced by high Spearman coefficients), validating our core hypothesis. Furthermore, we evaluate Dynamic Entropy Scheduling across diverse models and benchmarks. The results indicate that our method outperforms both the standard GRPO baseline and static regularization strategies, as well as other state-of-the-art RL optimization techniques.

### 6.1. Preliminaries: Group Relative Policy Optimization

We adopt **Group Relative Policy Optimization (GRPO)** from Shao et al. (2024) as our RL backbone. For each input query $x$ sampled from the dataset, the policy generates a group of $G$ completions $\{y_i\}_{i=1}^{G} \sim \pi_{\theta_{\mathrm{old}}}(\cdot \mid x)$, where $\pi_{\theta_{\mathrm{old}}}$ represents the policy parameters from the previous iteration. Let $r_i = r(x, y_i)$ denote the reward for the $i$-th completion. We compute the group-relative advantage by standardizing the rewards within the group: $A_i = [r_i - \mu(\{r_j\}_{j=1}^{G})]/\sigma(\{r_j\}_{j=1}^{G})$, where $\mu(\cdot)$ and $\sigma(\cdot)$ denote the empirical mean and standard deviation of the group rewards, respectively. GRPO optimizes a surrogate objective that incorporates the clipped PPO loss and a KL-divergence regularization term toward the reference policy $\pi_{\mathrm{ref}}$:

$$\mathcal{J}_{\mathrm{GRPO}}(\theta) = \mathbb{E}_x \mathbb{E}_{\{y_i\}_{i=1}^{G} \sim \pi_{\theta_{\mathrm{old}}}(\cdot|q)} \Big[ \frac{1}{G} \sum_{i=1}^{G} \frac{1}{|y_i|} \sum_{t=1}^{|y_i|}$$
$$\min\big\{ \rho_{i,t}(\theta) \, A_i, \ \mathrm{clip}\big(\rho_{i,t}(\theta), \, 1-\epsilon, \, 1+\epsilon\big) \, A_i \big\}$$
$$- \beta_{\mathrm{KL}} \mathbb{D}_{\mathrm{KL}}\big(\pi_\theta \,\big\|\, \pi^b\big) \Big], \qquad (3)$$

where $\beta_{\mathrm{KL}} > 0$ is a regularization parameter, and $\pi^b$ denotes the pretrained (base) policy.

### 6.2. Implementation of Entropy Discrepancy and Dynamic Entropy Scheduling

**Monte-Carlo estimation of entropy discrepancy.** Recall that Theorem 1 contrasts the entropy of the policy on *correct* versus *wrong* trajectories. In practice, during GRPO training

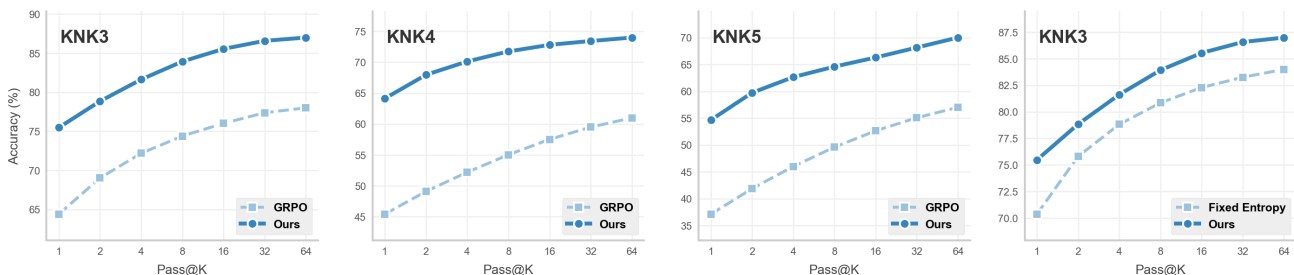

*Figure 2.* Comparison of dynamic scheduling and vanilla GRPO on Knights-and-Knaves puzzles; fixed entropy regularization is additionally shown on KNK-3 (right panel) as a reference.

we already sample a group of $G$ completions per input $x$, and we obtain a scalar reward $r(x, y_i)$ for each completion. We categorize each sampled completion into a correct set $\mathcal{C}$ and a wrong set $\mathcal{W}$ according to the verifiable rewards. Then for any sample $y_{i,t}$ with context $s_{i,t} = (x, y_{i,<t})$, the quantity $-\log \pi_\theta(y_{i,t} \mid s_{i,t})$ is an unbiased Monte-Carlo estimator of the conditional entropy $\mathcal{H}(\pi_\theta(\cdot \mid s_{i,t}))$ when $y_{i,t} \sim \pi_\theta(\cdot \mid s_{i,t})$.

---

**Estimation of Entropy Discrepancy**

We estimate the average entropy on correct and wrong trajectories by:

$$\widehat{H}_{\mathcal{W}}(\pi_\theta) = \frac{1}{N_{\mathcal{W}}} \sum_{(x,y)\in\mathcal{W}} \sum_{t=1}^{|y|} [-\log \pi_\theta(y_t|x, y_{<t})],$$

$$\widehat{H}_{\mathcal{C}}(\pi_\theta) = \frac{1}{N_{\mathcal{C}}} \sum_{(x,y)\in\mathcal{C}} \sum_{t=1}^{|y|} [-\log \pi_\theta(y_t|x, y_{<t})],$$

where parameters $N_{\mathcal{C}} = \sum_{(x,y)\in\mathcal{C}} |y|$ and $N_{\mathcal{W}} = \sum_{(x,y)\in\mathcal{W}} |y|$ are the total numbers in the correct and wrong sets, respectively. Then the entropy discrepancy is estimated as

$$\widehat{\text{Dis}}(\pi_\theta) = \widehat{H}_{\mathcal{C}}(\pi_\theta) - \widehat{H}_{\mathcal{W}}(\pi_\theta). \quad (4)$$

This estimator is directly available during RLVR without additional sampling.

---

**Dynamic entropy scheduling by discrepancy.** Building upon Theorem 3, we propose a dynamic entropy scheduling strategy to enhance RLVR. This approach adapts the entropy regularization strength based on the previously estimated discrepancy dynamics. The algorithm is formalized as follows:

---

**Dynamic entropy scheduling by discrepancy.**

We leverage the estimated Entropy Discrepancy $\widehat{\text{Dis}}(\pi_\theta)$ to dynamically modulate the entropy regularization strength. The objective function for our

---

dynamic scheduling is formulated as:

$$\mathcal{J}_{\text{RLVR}}(\theta) = \mathcal{J}_{\text{GRPO}}(\theta) + \gamma \cdot \widehat{\text{Dis}}(\pi_\theta)$$
$$\cdot \mathbb{E}_x \mathbb{E}_{y\sim\pi_\theta(\cdot|x)} \left[\frac{1}{|y|} \sum_{t=1}^{|y|} \mathcal{H}(\pi_\theta(\cdot \mid x, y_{<t}))\right],$$

where $\gamma$ is a constant hyperparameter scaling the regularization impact.

## 6.3. Correlation Between Entropy Discrepancy and Effect of Entropy Regularization

In this section, we provide empirical validation for the theoretical results presented in Sections 4 and 5. Specifically, we investigate the correlation between Entropy Discrepancy and the performance gains yielded by entropy regularization. Our analysis covers scenarios involving both increasing and decreasing entropy, as well as ablation studies on the regularization coefficient. Across all experimental settings, we observe an absolute Spearman correlation exceeding 0.8, indicating a robust relationship.

### 6.3.1. EXPERIMENTAL SETUP

We evaluate the correlation between Entropy Discrepancy and the performance impact of entropy regularization using the Countdown arithmetic reasoning task (Pan et al., 2025). In this task, the model must construct an equation using a specific set of integers to match a target value. To create a diverse set of experimental conditions, we vary the number and range of the input integers as well as the target value range. We conduct comparative experiments between the vanilla GRPO baseline and GRPO with fixed entropy coefficients across these distinct tasks. By plotting the Entropy Discrepancy against the performance difference (regularized vs. vanilla), we quantify the correlation between our metric and the regularization gain. Following Pan et al. (2025), we employ Qwen2.5-3B (Qwen Team, 2024) as the backbone model. Further experimental details are provided in Appendix C.

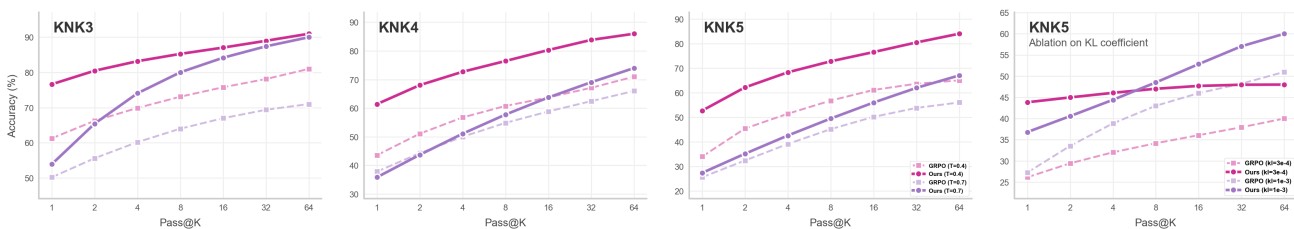

*Figure 3. Ablation Experiment on Temperature and KL Coefficient on Knights-and-Knaves (KNK) puzzle*

### 6.3.2. EXPERIMENTAL RESULTS

We now present our empirical evaluation. We begin by examining the correlation between Entropy Discrepancy and the effectiveness of positive entropy regularization.

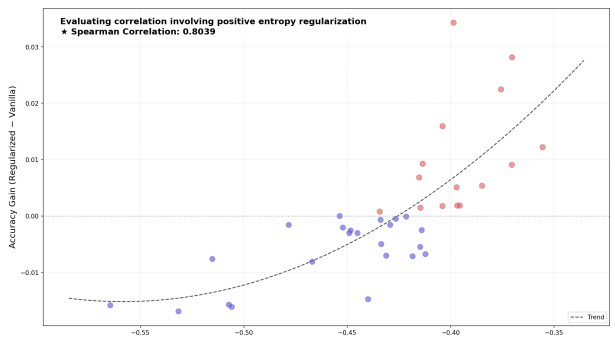

*Figure 4. Correlation involving positive entropy regularization.*

**Correlation involving positive entropy regularization.** We first investigate the relationship between Entropy Discrepancy and the benefits of positive entropy regularization. We fixed the regularization coefficient at $0.05$ and trained both vanilla GRPO and the regularized variant for 300 steps across 39 distinct tasks. We then evaluated the test accuracy for both methods. In Figure 4, we plot the performance gain yielded by positive entropy regularization against the Entropy Discrepancy. We observe a Spearman correlation of $0.8039$. Detailed task specifications are provided in Appendix C.

**Correlation involving negative entropy regularization.** Analogously, we evaluate the correlation between Entropy Discrepancy and the efficacy of negative entropy regularization. We set the coefficient to $-0.01$ and trained both vanilla GRPO and the regularized variant for 300 steps across 22 tasks. We then evaluated the test accuracy and plotted the performance gain against the Entropy Discrepancy. We observe a Spearman correlation of $-0.8729$, indicating a strong negative association. Results are shown in Figure 5.

**Ablation on entropy coefficient.** We further validate our observations by altering the entropy coefficient. Specifically, we adjusted the regularization coefficient to $0.01$ and compared the methods across 28 tasks over 300 training steps. Evaluating the relationship between performance gain and Entropy Discrepancy, we obtained a Spearman correlation of $0.8749$. This indicates that the strong positive correlation

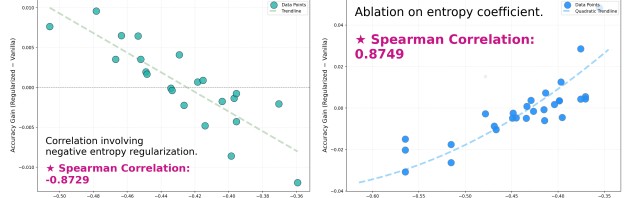

*Figure 5. Correlation involving negative entropy regularization (left), and ablation on entropy coefficient.*

remains consistent across different coefficient settings. See Figure 5 and Appendix C for details.

### 6.4. Dynamic Entropy Scheduling

In this section, we present the empirical evaluation of the dynamic entropy scheduling method proposed in Section 5. We assess our approach across two distinct tasks and model architectures. As illustrated in Figure 1, these tasks exhibit divergent entropy preferences: one (Knights-and-Knaves puzzle) benefits from increased entropy, while the other (math reasoning) favors decreased entropy to outperform vanilla RL. We demonstrate that our dynamic scheduling strategy adapts effectively to these conflicting requirements, achieving strong overall gains over vanilla GRPO on both accuracy-oriented and coverage-oriented metrics. Specifically, our method achieves up to **a 6.7 percentage-point Pass@K gain on mathematical reasoning** and **a 17.52 percentage-point gain on KNK5 at Pass@1**. Furthermore, ablation studies on sampling temperature reveal that our method achieves robust and broadly superior performance across the tested temperatures. Finally, we compare our approach with fixed entropy regularization and other recent methods designed to enhance RL reasoning, demonstrating favorable performance against these state-of-the-art baselines.

### 6.4.1. EXPERIMENT SETUP

We validate the effectiveness of discrepancy-based dynamic entropy scheduling across two domains: logical deduction and mathematical reasoning. We first consider the Knights-and-Knaves puzzle (Stojanovski et al., 2025), a classic logic problem involving inhabitants who either always tell the truth (Knights) or always lie (Knaves). The objective is to deduce the identity of each character based on a set of constrained statements. For this task, we train Qwen2.5-1.5B-Instruct (Qwen Team, 2024) on a mixed dataset comprising

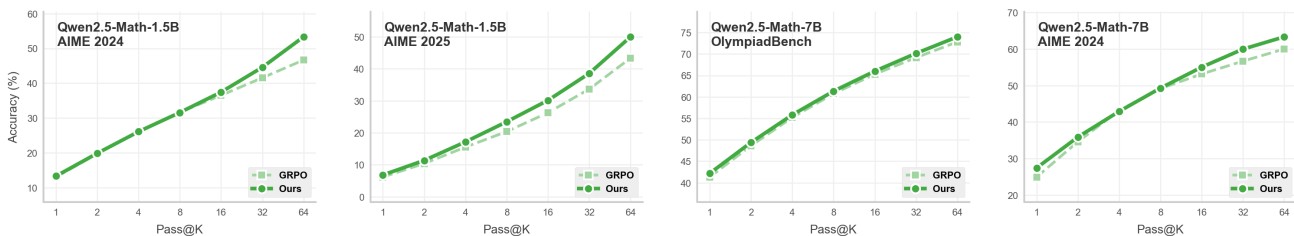

*Figure 6. Comparison of dynamic entropy scheduling and vanilla GRPO on math reasoning tasks*

puzzles of three difficulty levels (3–5) and evaluate it on strictly non-overlapping test sets.

For mathematical reasoning, we train Qwen2.5-Math-1.5B (Yang et al., 2024) and Qwen2.5-Math-7B (Yang et al., 2024) on the DAPO17k (Yu et al., 2025) dataset. We evaluate performance on four established benchmarks: MATH500 (Lightman et al., 2024), OlympiadBench (He et al., 2024), AIME24 (math ai, 2024), and AIME25 (math ai, 2025). Detailed experimental configurations and task descriptions are provided in Appendix D.

### 6.4.2. COMPARISON WITH VANILLA GRPO

We first compare dynamic entropy scheduling against the vanilla GRPO baseline. The results are presented in Figure 2 and Figure 6. On the Knights-and-Knaves (KNK) task, we observe that dynamic scheduling broadly improves over vanilla GRPO across the difficulty levels, with gains on both accuracy-oriented and coverage-oriented metrics. Specifically, our method achieves a 17.52 percentage-point gain over the GRPO baseline on KNK5 at Pass@1. Similarly, on mathematical reasoning benchmarks, dynamic entropy scheduling demonstrates strong overall performance across model scales and benchmark suites. Our proposed method broadly improves over vanilla GRPO and achieves a maximum Pass@K gain of 6.7 percentage points.

### 6.4.3. ABLATION ON EXPERIMENTS ON PARAMETERS

To further validate the effectiveness of dynamic scheduling, we conduct ablation experiments on hyperparameters. Specifically, we vary the sampling temperature and compare our method against static entropy regularization. Our results demonstrate that dynamic scheduling achieves robust performance across the tested hyperparameter settings and broadly improves over vanilla GRPO, confirming the effectiveness of adaptive entropy control.

**Ablation on sampling temperature.** We evaluate performance on the Knights-and-Knaves puzzles across three distinct sampling temperatures: $T \in \{0.4, 0.7, 1.0\}$. The results for $T = 1.0$ are presented in Figure 2, while the results for $T = 0.7$ and $T = 0.4$ are detailed in Figure 3. We observe that dynamic scheduling broadly improves over vanilla GRPO across the tested sampling temperatures.

**Ablation on the KL coefficient.** As noted by Liu et al.

(2025), adjusting the KL coefficient serves as an effective mechanism for entropy control. We conduct an ablation study by applying identical KL constraints to both our dynamic scheduling method and the GRPO baseline. Our results demonstrate that dynamic scheduling delivers strong gains over vanilla GRPO under both KL coefficients, with improvements averaging around 10 percentage points across the tested settings. This indicates that adaptive entropy control remains effective under different strengths of the KL constraint.

**Comparison with fixed entropy regularization.** We compare our method against fixed entropy regularization on both the Knights-and-Knaves puzzles and mathematical reasoning tasks. To ensure a rigorous comparison, we perform a grid search over the regularization coefficients and select the setting that yields the optimal performance. The results are visualized in Figure 2 and Figure 7. We observe that dynamic scheduling matches or surpasses the best-tuned fixed entropy regularization while avoiding the need to tune a separate fixed entropy coefficient.

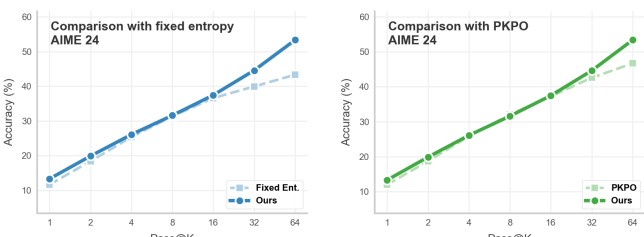

*Figure 7. Comparison with fixed entropy regularization (left) and comparison with PKPO (right)*

### 6.4.4. COMPARISON WITH OTHER METHODS

We further benchmark our proposed dynamic entropy scheduler against recent state-of-the-art methods, including Pass@K Optimization (PKPO) (Chen et al., 2025), Reward Unlikely (He et al., 2025), and Clipped IS-weight Policy Optimization (CISPO) (MiniMax et al., 2025). As illustrated in Figures 7 and 8, our method achieves performance parity or superiority over these specialized baselines. Notably, we achieve this efficacy while relying solely on a streamlined global entropy schedule, avoiding the complexity of intricate objective modifications.

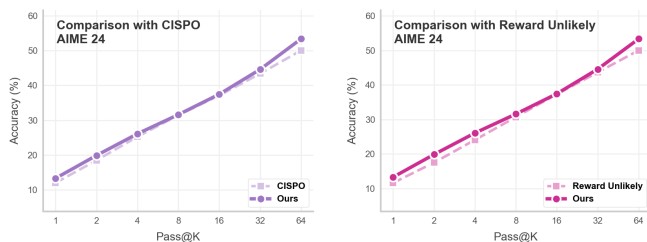

*Figure 8.* Comparison with CISPO (left) and comparison with Reward Unlikely (right).

---

**Takeaway: Empirical Findings**

1. **Validation of governing factors.** We demonstrate a strong correlation between Entropy Discrepancy and the performance gains yielded by entropy regularization (both positive and negative). Comprehensive ablation studies further corroborate the robustness of these results.

2. **Effectiveness of dynamic scheduling.** Our method outperforms vanilla RL across diverse tasks and models. Furthermore, it surpasses both fixed entropy regularization baselines and other state-of-the-art methods.

---

## 7. Conclusion

Previous literature presents conflicting findings regarding entropy regularization, with some studies advocating for increasing entropy while others report benefits from its reduction. In this paper, we reconcile these perspectives by demonstrating that the efficacy of entropy regularization is context-dependent. We establish a simplified yet empirically predictive theoretical model identifying Entropy Discrepancy as the governing factor: increasing entropy proves beneficial when the discrepancy is large, whereas decreasing entropy is optimal when the discrepancy is low. Building on this insight, we propose a dynamic entropy scheduling method that adaptively modulates the regularization coefficient. Our empirical evaluation is twofold: first, we confirm a strong correlation between Entropy Discrepancy and regularization gains, validating our theoretical model; second, we demonstrate that our proposed dynamic scheduling strategy significantly enhances RL performance across diverse tasks.

## Impact Statement

This paper presents work whose goal is to advance the understanding of entropy regularization in reinforcement learning for large language models. By clarifying when increasing versus decreasing entropy is appropriate and by providing a dynamic scheduling method, our contributions can make RL fine-tuning of LLMs more reliable and sample-efficient, which is beneficial for academic research as well as for practitioners. We do not introduce new datasets, new model checkpoints, or new deployment artifacts; all experiments use publicly released open-weight models (Qwen2.5, Qwen2.5-Math) and public benchmarks. As with other techniques that improve LLM reasoning capability, downstream applications inherit the well-known societal risks of LLMs (misuse, hallucination, biased outputs, etc.); our work neither materially exacerbates nor uniquely mitigates these risks, and we encourage practitioners to combine improved RL methods with appropriate safety, alignment, and evaluation practices.

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

# Appendix

## A. Additional Related Work

**Entropy control in reinforcement learning.** Entropy regularization serves as a fundamental mechanism in reinforcement learning to balance the exploration-exploitation trade-off. In the classic policy gradient framework, an entropy bonus is typically added to the objective function to prevent the policy from collapsing into a deterministic behavior too early (Williams, 1992; Mnih et al., 2016). This paradigm evolved into the Maximum Entropy Reinforcement Learning framework, where the agent aims to maximize both the expected return and the entropy of the policy (Ziebart et al., 2008; Haarnoja et al., 2018). Prominent algorithms like Soft Actor-Critic (SAC) (Haarnoja et al., 2018) and Soft Q-Learning (Haarnoja et al., 2017) explicitly treat entropy as a component of the value function, demonstrating that maintaining high entropy improves robustness and sample efficiency. While entropy maximization is the dominant strategy, dynamic adjustment techniques have also been explored. For instance, SAC introduces automated temperature tuning to constrain entropy to a target level, and other works employ entropy annealing schedules—gradually decaying the entropy coefficient to shift from exploration to exploitation as training progresses (Fox et al., 2015; Xu et al., 2025). However, these methods predominately view entropy as a quantity to be preserved or maximized for exploration, rather than minimized for reasoning stability.

**Theoretical analysis of entropy regularization** Beyond empirical success, the benefits of entropy regularization have been rigorously established through theoretical analysis. A primary line of research focuses on the optimization landscape. Ahmed et al. (2019) demonstrated that entropy regularization effectively smooths the non-convex optimization landscape, thereby preventing the policy from getting stuck in poor local minima. Furthermore, from the perspective of convergence rates, Mei et al. (2020) and Cen et al. (2022) provided proofs that entropy-regularized policy gradient methods achieve faster global convergence, often improving rates from sub-linear to linear. Additionally, Geist et al. (2019) and Neu et al. (2017) bridged entropy regularization with convex duality, interpreting it as a trust-region constraint that stabilizes policy updates. However, these theoretical guarantees are predominantly built upon the premise of entropy maximization. The theoretical properties of entropy minimization—or the dynamic interplay between increasing and decreasing entropy—remain largely under-explored in the existing literature.

**Entropy Regularization in LLM Fine-Tuning.** Entropy control is intricately linked to the stability of reinforcement learning from human/AI feedback (RLHF/RLAIF). Standard algorithms like PPO (Schulman et al., 2017) and GRPO (Shao et al., 2024) ostensibly manage policy updates via KL-divergence penalties. However, recent scrutiny reveals structural biases in these mechanisms. Park et al. (2025) demonstrate that the clipping mechanism in PPO and GRPO is not entropy-neutral; specifically, the "clip-high" effect implicitly drives entropy reduction independent of rewards, leading to premature mode collapse. Furthermore, Shen (2025) argue that traditional entropy bonuses are often ineffective in LLMs due to the immense action space and the sparsity of optimal tokens.

To mitigate these structural issues, recent works have proposed more sophisticated control mechanisms. Shen (2025) introduce a "clamped" entropy bonus to focus exploration on plausible tokens. Addressing the instability arising from positive-negative sample imbalance, Yang et al. (2025) apply Proportional-Integral (PI) control (EntroPIC) to stabilize entropy dynamics. Similarly, Kang et al. (2025) propose constraining entropy directly via output activation functions (ERA). While these methods improve stability, the optimal direction of entropy adjustment remains debated. While diversity is traditionally encouraged (Ouyang et al., 2022), recent empirical findings suggest that suppressing entropy (minimization) can enhance performance in complex reasoning tasks (Agarwal et al., 2025; Gao et al., 2025). Hybrid schedules combining monotonic or phase-based adjustments have also been explored (Zhang et al., 2025; Gai et al., 2026). Despite these advances, a unified theoretical framework explaining when to switch between entropy maximization and minimization remains absent. This work addresses this gap by deriving a principled criterion based on Entropy Discrepancy.

# B. Proofs of Theoretical Results

In this section, we provide the proofs for the theorems presented in Section 4 and Section 5. Following the parameterization in Section 3, all derivations are carried out for a single fixed prompt $x$: the shorthand $\pi(\tau)$ denotes the conditional policy $\pi_\theta(\mathbf{y} \mid x)$, every softmax normalization (and hence each $Z^{(t)}$) is taken over the responses $\mathbf{y} \in \mathcal{Y}$ to that prompt, and $\mathcal{C} \subseteq \mathcal{Y}$ is the set of correct responses for $x$.

This restriction to a single prompt is without loss of generality. Let $\mathcal{D}$ denote the distribution over prompts $x \in \mathcal{X}$, and write the population objective as $V(\pi) = \mathbb{E}_{x \sim \mathcal{D}}\big[\mathbb{E}_{\mathbf{y} \sim \pi(\cdot|x)}[r(x, \mathbf{y})]\big]$. Then the logits $\theta(x, \cdot)$ of distinct prompts share no coordinates, so the Fisher information matrix is block-diagonal across prompts, $\mathcal{F}_\theta = \mathrm{diag}_x\big(d(x)\,\mathcal{F}_\theta^x\big)$ with $d(x) = \mathcal{D}(x)$ and $\mathcal{F}_\theta^x$ the per-prompt Fisher block, while the Euclidean gradient block is $\nabla_{\theta(x,\cdot)} V = d(x)\,\nabla_{\theta(x,\cdot)}\big(\mathbb{E}_{\mathbf{y} \sim \pi(\cdot|x)}[r]\big)$. The prompt weight $d(x)$ therefore appears identically in the gradient and the Fisher block and cancels in the preconditioned update $\mathcal{F}_\theta^\dagger \nabla_\theta V$. The natural policy gradient update at each prompt thus depends only on that prompt's own policy and is independent of $\mathcal{D}$; the dynamics decouple across prompts, and the results below hold prompt-by-prompt.

## B.1. Proof for Theorem 2

> **Lemma 4.** *Consider the standard RL algorithm with the parameter update rule:*
>
> $$\theta_{van}^{(t+1)} = \theta_{van}^{(t)} + \eta \cdot \mathcal{F}_{\theta_{van}^{(t)}}^\dagger \nabla_\theta V(\pi^{(t)}),$$
>
> *The corresponding policy $\pi_{van}$ update takes the form:*
>
> $$\pi_{van}^{(t+1)}(\tau) \propto \pi_{van}^{(t)}(\tau) \exp(\eta \cdot r(\tau)).$$

*Proof.* We consider the softmax parameterization for the policy $\pi_{\text{van}}^{(t)}$, defined as:

$$\pi_{\text{van}}^{(t)}(\tau) = \frac{\exp(\theta_{\text{van}}^{(t)}(\tau))}{\sum_{\tau'} \exp(\theta_{\text{van}}^{(t)}(\tau'))}.$$

The value function is given by the expected reward $V(\pi_{\text{van}}^{(t)}) = \mathbb{E}_{\tau \sim \pi_{\text{van}}^{(t)}}[r(\tau)]$.

First, we derive the gradient of the value function with respect to the parameters $\theta$. Using the standard property of the softmax score function, $\nabla_{\theta(\tau)} \ln \pi_{\text{van}}^{(t)}(\tau') = \mathbb{I}(\tau = \tau') - \pi_{\text{van}}^{(t)}(\tau)$, the gradient is:

$$
\begin{aligned}
\nabla_{\theta(\tau)} V(\pi^{(t)}) &= \sum_{\tau'} r(\tau') \nabla_{\theta(\tau)} \pi_{\text{van}}^{(t)}(\tau') = \sum_{\tau'} r(\tau') \pi_{\text{van}}^{(t)}(\tau')\big(\mathbb{I}(\tau = \tau') - \pi_{\text{van}}^{(t)}(\tau)\big) \\
&= \pi^{(t)}(\tau)\big(r(\tau) - \sum_{\tau'} \pi_{\text{van}}^{(t)}(\tau')r(\tau')\big) \\
&= \pi_{\text{van}}^{(t)}(\tau)\big(r(\tau) - V(\pi_{\text{van}}^{(t)})\big).
\end{aligned}
\tag{5}
$$

Let $\mathbf{r}$ denote the vector of rewards. In vector notation, where $D_{\pi^{(t)}} = \mathrm{diag}(\pi^{(t)})$, this can be written as:

$$\nabla_\theta V(\pi_{\text{van}}^{(t)}) = D_{\pi_{\text{van}}^{(t)}}\big(\mathbf{r} - V(\pi_{\text{van}}^{(t)})\mathbf{1}\big).$$

Next, the Fisher Information Matrix (FIM) $\mathcal{F}_{\theta_{\text{van}}^{(t)}}$ for the softmax parameterization is:

$$\mathcal{F}_{\theta_{\text{van}}^{(t)}} = \mathbb{E}_{\tau \sim \pi_{\text{van}}^{(t)}}[\nabla \ln \pi_{\text{van}}^{(t)}(\tau) \nabla \ln \pi_{\text{van}}^{(t)}(\tau)^\top] = D_{\pi_{\text{van}}^{(t)}} - \pi_{\text{van}}^{(t)}(\pi_{\text{van}}^{(t)})^\top.$$

The NPG update direction $\Delta\theta$ is the solution to the linear system $\mathcal{F}_{\theta_{\text{van}}^{(t)}} \Delta\theta = \nabla_\theta V\big(\pi_{\text{van}}^{(t)}\big)$. We verify that the reward vector $\mathbf{r}$ is a valid solution. Multiplying $\mathcal{F}_{\theta_{\text{van}}^{(t)}}$ by $\mathbf{r}$:

$$\mathcal{F}_{\theta_{\text{van}}^{(t)}}\mathbf{r} = \big(D_{\pi_{\text{van}}^{(t)}} - \pi_{\text{van}}^{(t)}\big(\pi_{\text{van}}^{(t)}\big)^\top\big)\mathbf{r} = D_{\pi_{\text{van}}^{(t)}}\mathbf{r} - \pi_{\text{van}}^{(t)}\big(\big(\pi_{\text{van}}^{(t)}\big)^\top \mathbf{r}\big)$$

$$= \pi_{\text{van}}^{(t)} \odot \mathbf{r} - \pi_{\text{van}}^{(t)} V(\pi_{\text{van}}^{(t)}) \quad \text{(where } \odot \text{ denotes element-wise product)}$$

$$= D_{\pi_{\text{van}}^{(t)}}(\mathbf{r} - V(\pi_{\text{van}}^{(t)})\mathbf{1}).$$

Comparing this result with Eq. Equation (5), we observe that $\mathcal{F}_{\theta_{\text{van}}^{(t)}} \mathbf{r} = \nabla_\theta V(\pi_{\text{van}}^{(t)})$. Since the FIM is rank-deficient, the solution is unique up to a vector in the null space of $\mathcal{F}$ (constant vectors $c\mathbf{1}$). Due to the shift-invariance of the softmax function, any such solution yields the same policy. We choose the canonical solution $\mathcal{F}_{\theta^{(t)}}^\dagger \nabla_\theta V(\pi_{\text{van}}^{(t)}) = \mathbf{r}$.

Thus, the parameter update rule becomes:

$$\theta_{\text{van}}^{(t+1)}(\tau) = \theta_{\text{van}}^{(t)}(\tau) + \eta r(\tau).$$

Finally, we translate this parameter update to the policy space:

$$\pi_{\text{van}}^{(t+1)}(\tau) = \frac{\exp(\theta_{\text{van}}^{(t+1)}(\tau))}{\sum_{\tau'} \exp(\theta_{\text{van}}^{(t+1)}(\tau'))} = \frac{\exp(\theta_{\text{van}}^{(t)}(\tau) + \eta r(\tau))}{\sum_{\tau'} \exp(\theta_{\text{van}}^{(t+1)}(\tau'))} = \frac{\exp(\theta_{\text{van}}^{(t)}(\tau)) \exp(\eta r(\tau))}{Z^{(t+1)}},$$

where $Z^{(t+1)}$ is the normalization constant. Since $\pi^{(t)}(\tau) \propto \exp(\theta^{(t)}(\tau))$, we conclude:

$$\pi_{\text{van}}^{(t+1)}(\tau) \propto \pi_{\text{van}}^{(t)}(\tau) \exp(\eta r(\tau)).$$

$\square$

---

**Lemma 5.** *Consider the objectives for positive and negative entropy regularization, defined respectively as:*

$$V_+(\pi) = \mathbb{E}_{\tau \sim \pi}[r(\tau)] - \rho_+ \sum_\tau \pi(\tau) \ln \pi(\tau), \quad V_-(\pi) = \mathbb{E}_{\tau \sim \pi}[r(\tau)] + \rho_- \sum_\tau \pi(\tau) \ln \pi(\tau).$$

*Under the corresponding parameter update rules:*

$$\theta_+^{(t+1)} = \theta_+^{(t)} + \eta \mathcal{F}_{\theta_+^{(t)}}^\dagger \nabla_\theta V_+(\pi_+^{(t)}), \quad \theta_-^{(t+1)} = \theta_-^{(t)} + \eta \mathcal{F}_{\theta_-^{(t)}}^\dagger \nabla_\theta V_-(\pi_-^{(t)}),$$

*the policies form the following closed-form updates:*

$$\pi_+^{(t+1)}(\tau) \propto \left(\pi_+^{(t)}(\tau)\right)^{1-\eta\rho_+} \exp\left(\eta r(\tau)\right), \quad \pi_-^{(t+1)}(\tau) \propto \left(\pi_-^{(t)}(\tau)\right)^{1+\eta\rho_-} \exp\left(\eta r(\tau)\right).$$

---

*Proof.* We provide the proof for the positive entropy regularization case first, and then extend the result to the negative entropy regularization case.

**Case 1: Positive Entropy Regularization** $(V_+)$. Recall the objective function:

$$V_+(\pi) = \sum_\tau \pi(\tau) r(\tau) - \rho_+ \sum_\tau \pi(\tau) \ln \pi(\tau).$$

Let $\mathcal{H}(\pi) = -\sum_\tau \pi(\tau) \ln \pi(\tau)$ denote the entropy. We first compute the Euclidean gradient $\nabla_\theta V_+(\pi_+^{(t)})$. Let $g(\pi) = \sum_\tau \pi(\tau) \ln \pi(\tau)$. The gradient with respect to $\pi$ is $\nabla_\pi g(\pi) = \mathbf{1} + \ln \pi$. Using the chain rule and the Jacobian of the softmax transformation $J_\pi = \frac{\partial \pi}{\partial \theta} = D_\pi - \pi\pi^\top$, we have:

$$\nabla_\theta \left( \sum_\tau \pi(\tau) \ln \pi(\tau) \right) = (D_\pi - \pi\pi^\top)(\mathbf{1} + \ln \pi)$$
$$= (D_\pi \mathbf{1} - \pi\pi^\top \mathbf{1}) + (D_\pi \ln \pi - \pi\pi^\top \ln \pi)$$
$$= (\pi - \pi) + (D_\pi \ln \pi - \pi(-\mathcal{H}(\pi)))$$
$$= D_\pi(\ln \pi + \mathcal{H}(\pi)\mathbf{1}).$$

Combining this with the gradient of the expected reward derived in Lemma 4 (which is $D_\pi(\mathbf{r} - \mathbb{E}[r]\mathbf{1})$), the total gradient is:

$$\nabla_\theta V_+(\pi_+^{(t)}) = D_{\pi_+^{(t)}}(\mathbf{r} - \mathbb{E}_{\pi_+^{(t)}}[r]\mathbf{1}) - \rho_+ D_{\pi_+^{(t)}}(\ln \pi_+^{(t)} + \mathcal{H}(\pi_+^{(t)})\mathbf{1})$$

$$= D_{\pi_+^{(t)}}\left(\mathbf{r} - \rho_+ \ln \pi_+^{(t)} - (\mathbb{E}_{\pi_+^{(t)}}[r] + \rho_+ \mathcal{H}(\pi_+^{(t)}))\mathbf{1}\right)$$

$$= D_{\pi_+^{(t)}}\left(\mathbf{r} - \rho_+ \ln \pi_+^{(t)} - V_+(\pi_+^{(t)})\mathbf{1}\right). \tag{6}$$

We verify that the natural gradient direction $\Delta\theta_+ = \mathbf{r} - \rho_+ \ln \pi_+^{(t)}$ satisfies the system $\mathcal{F}_{\theta_+^{(t)}}\Delta\theta_+ = \nabla_\theta V_+(\pi_+^{(t)})$.

$$\begin{aligned}
\mathcal{F}_{\theta_+^{(t)}}(\mathbf{r} - \rho_+ \ln \pi_+^{(t)}) &= (D_{\pi_+^{(t)}} - \pi_+^{(t)}(\pi_+^{(t)})^\top)(\mathbf{r} - \rho_+ \ln \pi_+^{(t)}) \\
&= [D_{\pi_+^{(t)}}\mathbf{r} - \pi_+^{(t)}\mathbb{E}[r]] - \rho_+[D_{\pi_+^{(t)}}\ln \pi_+^{(t)} - \pi_+^{(t)}(-\mathcal{H}(\pi_+^{(t)}))] \\
&= D_{\pi_+^{(t)}}(\mathbf{r} - \rho_+ \ln \pi_+^{(t)}) - \pi_+^{(t)}(\mathbb{E}[r] + \rho_+ \mathcal{H}(\pi_+^{(t)})) \\
&= D_{\pi_+^{(t)}}(\mathbf{r} - \rho_+ \ln \pi_+^{(t)}) - \pi_+^{(t)}V_+(\pi_+^{(t)}) \\
&= D_{\pi_+^{(t)}}(\mathbf{r} - \rho_+ \ln \pi_+^{(t)} - V_+(\pi_+^{(t)})\mathbf{1}).
\end{aligned}$$

This matches Eq. Equation (6). Thus, the parameter update is:

$$\theta_+^{(t+1)}(\tau) = \theta_+^{(t)}(\tau) + \eta(r(\tau) - \rho_+ \ln \pi_+^{(t)}(\tau)).$$

Recalling that $\ln \pi_+^{(t)}(\tau) = \theta_+^{(t)}(\tau) - \ln Z_+^{(t)}$, we substitute this into the update rule:

$$\theta_+^{(t+1)}(\tau) = \theta_+^{(t)}(\tau) + \eta r(\tau) - \eta\rho_+(\theta_+^{(t)}(\tau) - \ln Z_+^{(t)}) = (1 - \eta\rho_+)\theta_+^{(t)}(\tau) + \eta r(\tau) + \eta\rho_+ \ln Z_+^{(t)}.$$

Exponentiating to find the policy update:

$$\begin{aligned}
\pi_+^{(t+1)}(\tau) &\propto \exp(\theta_+^{(t+1)}(\tau)) \propto \exp\left((1 - \eta\rho_+)\theta_+^{(t)}(\tau) + \eta r(\tau)\right) \propto \left(\exp(\theta_+^{(t)}(\tau))\right)^{1-\eta\rho_+}\exp(\eta r(\tau)) \\
&\propto (\pi_+^{(t)}(\tau))^{1-\eta\rho_+}\exp(\eta r(\tau)).
\end{aligned}$$

**Case 2: Negative Entropy Regularization ($V_-$).** The objective is $V_-(\pi) = \mathbb{E}[r] + \rho_- \sum \pi \ln \pi$. This is algebraically equivalent to the Case 1 with $\rho_+$ replaced by $-\rho_-$. Following the same derivation steps, the natural gradient direction becomes $\Delta\theta_- = \mathbf{r} + \rho_- \ln \pi_-^{(t)}$. The parameter update is:

$$\begin{aligned}
\theta_-^{(t+1)}(\tau) &= \theta_-^{(t)}(\tau) + \eta(r(\tau) + \rho_- \ln \pi_-^{(t)}(\tau)) \\
&= \theta_-^{(t)}(\tau) + \eta r(\tau) + \eta\rho_-(\theta_-^{(t)}(\tau) - \ln Z_-^{(t)}) \\
&= (1 + \eta\rho_-)\theta_-^{(t)}(\tau) + \eta r(\tau) - \eta\rho_- \ln Z_-^{(t)}.
\end{aligned}$$

Translating to the policy space:

$$\pi_-^{(t+1)}(\tau) \propto \exp\left((1 + \eta\rho_-)\theta_-^{(t)}(\tau) + \eta r(\tau)\right) \propto (\pi_-^{(t)}(\tau))^{1+\eta\rho_-}\exp(\eta r(\tau)).$$

This concludes the proof. □

**Definition 6.** *We define the correctness score $C(\pi)$ as the log-odds of the policy generating a correct solution. Formally, let $\mathcal{C}$ be the set of correct trajectories; we define:*

$$C(\pi) = \ln\left(\frac{\sum_{\tau \in \mathcal{C}} \pi(\tau)}{\sum_{\tau \notin \mathcal{C}} \pi(\tau)}\right).$$

We now proceed to analyze the evolution of $C(\pi^{(T)})$ under the different RL algorithms proposed above.

**Lemma 7.** *Consider the Reinforcement Learning setting with negative entropy regularization parameterized by coefficient $\rho_-$. Let $C(\pi) \triangleq \ln\left(\frac{\sum_{\tau \in \mathcal{C}} \pi(\tau)}{\sum_{\tau \notin \mathcal{C}} \pi(\tau)}\right)$ be the log-odds of the policy mass on the target set $\mathcal{C}$. Assuming binary rewards where $r(\tau) = 1$ if $\tau \in \mathcal{C}$ and $r(\tau) = 0$ otherwise, for any training step $T$, the sensitivity $\frac{\partial C(\pi_-^{(T)})}{\partial \rho_-}\big|_{\rho_-=0} > 0$ holds if and only if:*

$$\frac{1}{2}\eta(T-1) + \left[\frac{\sum_{\tau \in \mathcal{C}} \pi^{(0)}(\tau) \ln \pi^{(0)}(\tau)}{\sum_{\tau \in \mathcal{C}} \pi^{(0)}(\tau)} - \frac{\sum_{\tau \notin \mathcal{C}} \pi^{(0)}(\tau) \ln \pi^{(0)}(\tau)}{\sum_{\tau \notin \mathcal{C}} \pi^{(0)}(\tau)}\right] > 0. \tag{7}$$

*Proof.* Let $C^{(t)}$ denote $C(\pi^{(t)})$. The policy update rule with negative entropy regularization is given by $\pi^{(t+1)}(\tau) \propto (\pi^{(t)}(\tau))^{1+\eta\rho_-} \exp(\eta r(\tau))$. We analyze the recurrence of the derivative with respect to $\rho_-$ evaluated at $\rho_- = 0$. The metric $C^{(t+1)}$ is defined as:

$$C^{(t+1)} = \ln\left(\sum_{\tau \in \mathcal{C}} (\pi^{(t)}(\tau))^{1+\eta\rho_-} \exp(\eta)\right) - \ln\left(\sum_{\tau \notin \mathcal{C}} (\pi^{(t)}(\tau))^{1+\eta\rho_-}\right).$$

Differentiating with respect to $\rho_-$ and setting $\rho_- = 0$ (where the update degenerates to the standard NPG update $\pi^{(t+1)}(\tau) \propto \pi^{(t)}(\tau) \exp(\eta r(\tau))$):

$$\frac{\partial C^{(t+1)}}{\partial \rho_-}\bigg|_{\rho_-=0} = \frac{\sum_{\tau \in \mathcal{C}} \pi^{(t+1)}(\tau)\left[\frac{1}{\pi^{(t)}(\tau)}\frac{\partial \pi^{(t)}(\tau)}{\partial \rho_-} + \eta \ln \pi^{(t)}(\tau)\right]}{\sum_{\tau \in \mathcal{C}} \pi^{(t+1)}(\tau)} - \frac{\sum_{\tau \notin \mathcal{C}} \pi^{(t+1)}(\tau)\left[\frac{1}{\pi^{(t)}(\tau)}\frac{\partial \pi^{(t)}(\tau)}{\partial \rho_-} + \eta \ln \pi^{(t)}(\tau)\right]}{\sum_{\tau \notin \mathcal{C}} \pi^{(t+1)}(\tau)}.$$

Under the vanilla trajectory ($\rho_- = 0$), the conditional probabilities within the sets $\mathcal{C}$ and $\mathcal{C}^c$ remain invariant because the reward is constant within each group. That is, $\frac{\pi^{(t+1)}(\tau)}{\sum_{\tau' \in \mathcal{C}} \pi^{(t+1)}(\tau')} = \frac{\pi^{(t)}(\tau)}{\sum_{\tau' \in \mathcal{C}} \pi^{(t)}(\tau')}$. Consequently, the terms involving $\frac{\partial \pi^{(t)}}{\partial \rho_-}$ collapse into $\frac{\partial C^{(t)}}{\partial \rho_-}$. The recurrence relation simplifies to:

$$\frac{\partial C^{(t+1)}}{\partial \rho_-}\bigg|_{\rho_-=0} = \frac{\partial C^{(t)}}{\partial \rho_-}\bigg|_{\rho_-=0} + \eta \underbrace{\left[\frac{\sum_{\tau \in \mathcal{C}} \pi^{(t)}(\tau) \ln \pi^{(t)}(\tau)}{\sum_{\tau \in \mathcal{C}} \pi^{(t)}(\tau)} - \frac{\sum_{\tau \notin \mathcal{C}} \pi^{(t)}(\tau) \ln \pi^{(t)}(\tau)}{\sum_{\tau \notin \mathcal{C}} \pi^{(t)}(\tau)}\right]}_{\Delta H^{(t)}}.$$

We now evaluate the term $\Delta H^{(t)}$. For the vanilla update (where $\rho_- = 0$), the policy evolves as $\pi^{(t)}(\tau) \propto \pi^{(0)}(\tau) \exp(\eta t \cdot r(\tau))$. For $\tau \in \mathcal{C}$, $\ln \pi^{(t)}(\tau) = \ln \pi^{(0)}(\tau) + \eta t - \ln Z_t$. For $\tau \notin \mathcal{C}$, $\ln \pi^{(t)}(\tau) = \ln \pi^{(0)}(\tau) - \ln Z_t$. Substituting these into $\Delta H^{(t)}$, the normalization term $\ln Z_t$ cancels out between the two groups. The term $\eta t$ appears only in the first group (target set $\mathcal{C}$). Thus:

$$\Delta H^{(t)} = \Delta H^{(0)} + \eta t.$$

Summing the recurrence from $t = 0$ to $T-1$ (with $\frac{\partial C^{(0)}}{\partial \rho_-} = 0$):

$$\frac{\partial C^{(T)}}{\partial \rho_-}\bigg|_{\rho_-=0} = \eta \sum_{t=0}^{T-1}(\Delta H^{(0)} + \eta t) = \eta T \Delta H^{(0)} + \eta^2 \frac{T(T-1)}{2}.$$

The condition $\frac{\partial C^{(T)}}{\partial \rho_-} > 0$ is thus equivalent to:

$$\eta T \Delta H^{(0)} + \frac{1}{2}\eta^2 T(T-1) > 0 \quad \Longleftrightarrow \quad \Delta H^{(0)} + \frac{1}{2}\eta(T-1) > 0.$$

Substituting the definition of $\Delta H^{(0)}$ yields the stated condition. $\qquad \square$

**Lemma 8.** *Consider the Reinforcement Learning setting with positive entropy regularization parameterized by coefficient $\rho_+$. Let $C(\pi) \triangleq \ln\left(\frac{\sum_{\tau \in \mathcal{C}} \pi(\tau)}{\sum_{\tau \notin \mathcal{C}} \pi(\tau)}\right)$ be the log-odds of the policy mass on the target set $\mathcal{C}$. Assuming binary rewards where $r(\tau) = 1$ if $\tau \in \mathcal{C}$ and $r(\tau) = 0$ otherwise, for any training step $T$, the sensitivity $\frac{\partial C(\pi_+^{(T)})}{\partial \rho_+}\big|_{\rho_+=0} > 0$ holds if and only if:*

$$\frac{1}{2}\eta(T-1) + \left[\frac{\sum_{\tau \in \mathcal{C}} \pi^{(0)}(\tau) \ln \pi^{(0)}(\tau)}{\sum_{\tau \in \mathcal{C}} \pi^{(0)}(\tau)} - \frac{\sum_{\tau \notin \mathcal{C}} \pi^{(0)}(\tau) \ln \pi^{(0)}(\tau)}{\sum_{\tau \notin \mathcal{C}} \pi^{(0)}(\tau)}\right] < 0. \tag{8}$$

*Proof.* Let $C^{(t)}$ denote $C(\pi_+^{(t)})$. The policy update rule with positive entropy regularization is given by $\pi^{(t+1)}(\tau) \propto (\pi^{(t)}(\tau))^{1-\eta\rho_+} \exp(\eta r(\tau))$. We analyze the recurrence of the derivative with respect to $\rho_+$ evaluated at $\rho_+ = 0$. The metric $C^{(t+1)}$ is explicitly defined as:

$$C^{(t+1)} = \ln\left(\sum_{\tau \in \mathcal{C}} (\pi^{(t)}(\tau))^{1-\eta\rho_+} \exp(\eta)\right) - \ln\left(\sum_{\tau \notin \mathcal{C}} (\pi^{(t)}(\tau))^{1-\eta\rho_+}\right).$$

Differentiating with respect to $\rho_+$ and evaluating at $\rho_+ = 0$ (where the update degenerates to the standard NPG update $\pi^{(t+1)}(\tau) \propto \pi^{(t)}(\tau) \exp(\eta r(\tau))$):

$$\frac{\partial C^{(t+1)}}{\partial \rho_+}\bigg|_{\rho_+=0} = \frac{\sum_{\tau \in \mathcal{C}} \pi^{(t+1)}(\tau)\left[\frac{1}{\pi^{(t)}(\tau)}\frac{\partial \pi^{(t)}(\tau)}{\partial \rho_+} - \eta \ln \pi^{(t)}(\tau)\right]}{\sum_{\tau \in \mathcal{C}} \pi^{(t+1)}(\tau)} - \frac{\sum_{\tau \notin \mathcal{C}} \pi^{(t+1)}(\tau)\left[\frac{1}{\pi^{(t)}(\tau)}\frac{\partial \pi^{(t)}(\tau)}{\partial \rho_+} - \eta \ln \pi^{(t)}(\tau)\right]}{\sum_{\tau \notin \mathcal{C}} \pi^{(t+1)}(\tau)}.$$

Note that the term $-\eta \ln \pi^{(t)}(\tau)$ arises from the derivative of the exponent $1 - \eta\rho_+$. Under the vanilla trajectory ($\rho_+ = 0$), the conditional probabilities within the sets $\mathcal{C}$ and $\mathcal{C}^c$ remain invariant because the reward is constant within each group (binary reward assumption). That is, $\frac{\pi^{(t+1)}(\tau)}{\sum_{\tau' \in \mathcal{C}} \pi^{(t+1)}(\tau')} = \frac{\pi^{(t)}(\tau)}{\sum_{\tau' \in \mathcal{C}} \pi^{(t)}(\tau')}$. Consequently, the terms involving $\frac{\partial \pi^{(t)}}{\partial \rho_+}$ collapse into $\frac{\partial C^{(t)}}{\partial \rho_+}$. The recurrence relation simplifies to:

$$\frac{\partial C^{(t+1)}}{\partial \rho_+}\bigg|_{\rho_+=0} = \frac{\partial C^{(t)}}{\partial \rho_+}\bigg|_{\rho_+=0} - \eta \underbrace{\left[\frac{\sum_{\tau \in \mathcal{C}} \pi^{(t)}(\tau) \ln \pi^{(t)}(\tau)}{\sum_{\tau \in \mathcal{C}} \pi^{(t)}(\tau)} - \frac{\sum_{\tau \notin \mathcal{C}} \pi^{(t)}(\tau) \ln \pi^{(t)}(\tau)}{\sum_{\tau \notin \mathcal{C}} \pi^{(t)}(\tau)}\right]}_{\Delta H^{(t)}}.$$

We now evaluate the term $\Delta H^{(t)}$. For the vanilla update (where $\rho_+ = 0$), the policy evolves as $\pi^{(t)}(\tau) \propto \pi^{(0)}(\tau) \exp(\eta t \cdot r(\tau))$. For $\tau \in \mathcal{C}$, $\ln \pi^{(t)}(\tau) = \ln \pi^{(0)}(\tau) + \eta t - \ln Z_t$. For $\tau \notin \mathcal{C}$, $\ln \pi^{(t)}(\tau) = \ln \pi^{(0)}(\tau) - \ln Z_t$. Substituting these into $\Delta H^{(t)}$, the normalization term $\ln Z_t$ cancels out between the two groups. The term $\eta t$ appears only in the first group (target set $\mathcal{C}$). Thus:

$$\Delta H^{(t)} = \Delta H^{(0)} + \eta t.$$

Summing the recurrence from $t = 0$ to $T - 1$ (noting $\frac{\partial C^{(0)}}{\partial \rho_+} = 0$):

$$\frac{\partial C^{(T)}}{\partial \rho_+}\bigg|_{\rho_+=0} = -\eta \sum_{t=0}^{T-1} (\Delta H^{(0)} + \eta t) = -\eta T \Delta H^{(0)} - \eta^2 \frac{T(T-1)}{2} = -\eta T \left(\Delta H^{(0)} + \frac{1}{2}\eta(T-1)\right).$$

The condition $\frac{\partial C^{(T)}}{\partial \rho_+} > 0$ requires the term inside the parenthesis to be negative (since $-\eta T < 0$):

$$\Delta H^{(0)} + \frac{1}{2}\eta(T-1) < 0.$$

Substituting the definition of $\Delta H^{(0)}$ yields the stated condition. $\qquad\square$

**Theorem 9.** *There exists a regularization coefficient $\rho_- > 0$ such that negative entropy regularization outperforms vanilla RL, i.e., $V(\pi_-^{(T)}) > V(\pi_{van}^{(T)})$, provided that the following condition holds:*

$$\frac{1}{2}\eta(T-1) + \left[\frac{\sum_{\tau \in \mathcal{C}} \pi_{base}(\tau) \ln \pi_{base}(\tau)}{\sum_{\tau \in \mathcal{C}} \pi_{base}(\tau)} - \frac{\sum_{\tau \notin \mathcal{C}} \pi_{base}(\tau) \ln \pi_{base}(\tau)}{\sum_{\tau \notin \mathcal{C}} \pi_{base}(\tau)}\right] > 0. \tag{9}$$

*Conversely, if the inequality in Equation (9) does not hold (i.e., the expression is non-positive), there exists a coefficient $\rho_+ > 0$ such that positive entropy regularization yields a higher value, $V(\pi_+^{(T)}) > V(\pi_{van}^{(T)})$.*

*Proof.* Invoking Lemma 8 and Lemma 7, we establish that when Inequality Equation (9) holds, the sensitivity of the correctness metric with respect to the negative regularization coefficient is positive:

$$\left.\frac{\partial C(\pi_-^{(T)})}{\partial \rho_-}\right|_{\rho_-=0} > 0.$$

This implies the existence of a regularization coefficient $\rho_- > 0$ such that negative entropy regularization outperforms the vanilla baseline. Similarly, if Inequality Equation (9) does not hold, we obtain $\left.\frac{\partial C(\pi_+^{(T)})}{\partial \rho_+}\right|_{\rho_+=0} > 0$, guaranteeing the existence of a coefficient $\rho_+ > 0$ such that positive entropy regularization yields a superior value, $V(\pi_+^{(T)}) > V(\pi_{van}^{(T)})$. $\square$

### B.2. Proof for Theorem 3

**Lemma 10.** *Consider the dynamic entropy scheduling with the parameter update rule:*

$$\theta_{dy}^{(t+1)} = \theta_{dy}^{(t)} + \eta \cdot \mathcal{F}_{\theta_{dy}^{(t)}}^\dagger \nabla_\theta V_{dy}(\pi_{dy}^{(t)}),$$

*where for any policy $\pi$ the objective $V_{dy}(\pi)$ is defined as*

$$V_{dy}(\pi) = \mathbb{E}_{\tau \sim \pi}[r(\tau)] - \rho_{dy}\text{Dis}(\pi_{no\ grad}) \cdot \sum_\tau \pi(\tau) \ln \pi(\tau),$$

*The corresponding policy $\pi_{van}$ update takes the form:*

$$\pi_{dy}^{(t+1)}(\tau) \propto \left(\pi_{dy}^{(t)}(\tau)\right)^{1-\eta \cdot \rho_{dy} \cdot \text{Dis}(\pi_{dy}^{(t)})} \exp(\eta \cdot r(\tau)).$$

*Proof.* Let $\pi_{dy}^{(t)}$ denote the policy at iteration $t$. We simplify the notation for the regularization coefficient at step $t$ as $\Lambda_t$. Since the term $\text{Dis}(\pi_{no\ grad})$ indicates a stop-gradient operation (treating the distribution metric as a fixed scalar during optimization at step $t$), we define:

$$\Lambda_t \triangleq \rho_{dy} \cdot \text{Dis}(\pi_{dy}^{(t)}). \tag{10}$$

Thus, at iteration $t$, the objective function for the purpose of gradient computation is effectively:

$$V_{dy}(\pi) \cong \mathbb{E}_{\tau \sim \pi}[r(\tau)] - \Lambda_t \sum_\tau \pi(\tau) \ln \pi(\tau). \tag{11}$$

**Step 1: Gradient Computation.** We compute the gradient $\nabla_\theta V_{dy}(\pi_{dy}^{(t)})$. The gradient of the expected reward is standard: $\nabla_\theta \mathbb{E}[r] = (D_\pi - \pi\pi^\top)\mathbf{r}$. For the entropy term, noting that $\Lambda_t$ is treated as a constant w.r.t. $\theta$, the gradient w.r.t. $\pi$ is $-\Lambda_t(\mathbf{1} + \ln \pi)$. Applying the chain rule with the Jacobian $D_\pi - \pi\pi^\top$:

$$\nabla_\theta\left(-\Lambda_t \sum_\tau \pi(\tau) \ln \pi(\tau)\right) = -\Lambda_t(D_\pi - \pi\pi^\top)(\mathbf{1} + \ln \pi) = -\Lambda_t(D_\pi - \pi\pi^\top)\ln \pi. \tag{12}$$

Note that $(D_\pi - \pi\pi^\top)\mathbf{1} = \mathbf{0}$, eliminating the constant component of the derivative. Combining terms, the total gradient is:

$$\nabla_\theta V_{dy}(\pi_{dy}^{(t)}) = \mathcal{F}_{\theta_{dy}^{(t)}}(\mathbf{r} - \Lambda_t \ln \pi_{dy}^{(t)}), \tag{13}$$

where $\mathcal{F}_\theta = D_\pi - \pi\pi^\top$ is the Fisher Information Matrix.

**Step 2: Natural Gradient Update.** The update rule is $\theta_{\text{dy}}^{(t+1)} = \theta_{\text{dy}}^{(t)} + \eta \mathcal{F}^{\dagger} \nabla V$. Solving the linear system $\mathcal{F} \Delta \theta = \mathcal{F}(\mathbf{r} - \Lambda_t \ln \pi_{\text{dy}}^{(t)})$, we identify the natural gradient direction (ignoring shift-invariant baseline terms) as:

$$\Delta \theta(\tau) = r(\tau) - \Lambda_t \ln \pi_{\text{dy}}^{(t)}(\tau). \tag{14}$$

Substituting this into the update rule:

$$\theta_{\text{dy}}^{(t+1)}(\tau) = \theta_{\text{dy}}^{(t)}(\tau) + \eta r(\tau) - \eta \Lambda_t \ln \pi_{\text{dy}}^{(t)}(\tau). \tag{15}$$

Using the relation $\ln \pi_{\text{dy}}^{(t)}(\tau) = \theta_{\text{dy}}^{(t)}(\tau) - \ln Z^{(t)}$, where $Z^{(t)}$ is the normalization constant:

$$\theta_{\text{dy}}^{(t+1)}(\tau) = \theta_{\text{dy}}^{(t)}(\tau) + \eta r(\tau) - \eta \Lambda_t(\theta_{\text{dy}}^{(t)}(\tau) - \ln Z^{(t)}) = (1 - \eta \Lambda_t)\theta_{\text{dy}}^{(t)}(\tau) + \eta r(\tau) + \eta \Lambda_t \ln Z^{(t)}. \tag{16}$$

**Step 3: Policy Update Form.** Exponentiating the parameters to obtain the policy update:

$$\pi_{\text{dy}}^{(t+1)}(\tau) \propto \exp\left(\theta_{\text{dy}}^{(t+1)}(\tau)\right) \propto \exp\left((1 - \eta \Lambda_t)\theta_{\text{dy}}^{(t)}(\tau) + \eta r(\tau)\right)$$

$$\propto \left(\exp(\theta_{\text{dy}}^{(t)}(\tau))\right)^{1 - \eta \Lambda_t} \exp(\eta r(\tau))$$

$$\propto (\pi_{\text{dy}}^{(t)}(\tau))^{1 - \eta \Lambda_t} \exp(\eta r(\tau)).$$

Substituting $\Lambda_t = \rho_{\text{dy}} \text{Dis}(\pi_{\text{dy}}^{(t)})$ back into the expression completes the proof. $\square$

---

**Lemma 11.** *Consider the dynamic entropy scheduling with coefficient $\rho_{dy}$. Let $C(\pi) \triangleq \ln\left(\frac{\sum_{\tau \in \mathcal{C}} \pi(\tau)}{\sum_{\tau \notin \mathcal{C}} \pi(\tau)}\right)$ denote the log-odds of the policy mass on the target set $\mathcal{C}$. We define the discrepancy metric $\text{Dis}(\pi)$ as the difference in the average log-likelihood between the target set $\mathcal{C}$ and its complement $\mathcal{C}^c$:*

$$\text{Dis}(\pi) \triangleq \frac{\sum_{\tau \notin \mathcal{C}} \pi(\tau) \ln \pi(\tau)}{\sum_{\tau \notin \mathcal{C}} \pi(\tau)} - \frac{\sum_{\tau \in \mathcal{C}} \pi(\tau) \ln \pi(\tau)}{\sum_{\tau \in \mathcal{C}} \pi(\tau)}. \tag{17}$$

*Assuming binary rewards where $r(\tau) = 1$ if $\tau \in \mathcal{C}$ and $r(\tau) = 0$ otherwise, for any training step $T$, the sensitivity is strictly positive:*

$$\left.\frac{\partial C(\pi_{dy}^{(T)})}{\partial \rho_{dy}}\right|_{\rho_{dy}=0} > 0. \tag{18}$$

---

*Proof.* Let $C^{(t)}$ denote $C(\pi_{\text{dy}}^{(t)})$. The update rule under dynamic entropy scheduling is given by:

$$\pi^{(t+1)}(\tau) \propto (\pi^{(t)}(\tau))^{1 - \eta \rho_{\text{dy}} \text{Dis}(\pi_{\text{ng}}^{(t)})} \exp(\eta r(\tau)), \tag{19}$$

where $\pi_{\text{ng}}^{(t)}$ denotes the policy with a stop-gradient operator (i.e., $\text{Dis}(\pi^{(t)})$ is treated as a constant scalar $\Lambda_t$ during differentiation w.r.t. $\rho_{\text{dy}}$).

We analyze the recurrence of the gradient. Expanding the definition of $C^{(t+1)}$:

$$C^{(t+1)} = \ln\left(\sum_{\tau \in \mathcal{C}}(\pi^{(t)}(\tau))^{1 - \eta \rho_{\text{dy}} \Lambda_t} \exp(\eta)\right) - \ln\left(\sum_{\tau \notin \mathcal{C}}(\pi^{(t)}(\tau))^{1 - \eta \rho_{\text{dy}} \Lambda_t}\right).$$

Differentiating with respect to $\rho_{\text{dy}}$ and evaluating at $\rho_{\text{dy}} = 0$:

$$\left.\frac{\partial C^{(t+1)}}{\partial \rho_{\text{dy}}}\right|_{\rho_{\text{dy}}=0} = \frac{\sum_{\tau \in \mathcal{C}} \pi^{(t+1)}(\tau)\left[\frac{1}{\pi^{(t)}(\tau)}\frac{\partial \pi^{(t)}(\tau)}{\partial \rho_{\text{dy}}} - \eta \Lambda_t \ln \pi^{(t)}(\tau)\right]}{\sum_{\tau \in \mathcal{C}} \pi^{(t+1)}(\tau)} - \frac{\sum_{\tau \notin \mathcal{C}} \pi^{(t+1)}(\tau)\left[\frac{1}{\pi^{(t)}(\tau)}\frac{\partial \pi^{(t)}(\tau)}{\partial \rho_{\text{dy}}} - \eta \Lambda_t \ln \pi^{(t)}(\tau)\right]}{\sum_{\tau \notin \mathcal{C}} \pi^{(t+1)}(\tau)}.$$

At $\rho_{\text{dy}} = 0$, the conditional probabilities within each group are invariant (due to the binary reward structure), allowing the recursive terms $\frac{\partial \pi^{(t)}}{\partial \rho_{\text{dy}}}$ to collapse into $\frac{\partial C^{(t)}}{\partial \rho_{\text{dy}}}$. The equation simplifies to:

$$\left.\frac{\partial C^{(t+1)}}{\partial \rho_{\text{dy}}}\right|_{\rho_{\text{dy}}=0} = \left.\frac{\partial C^{(t)}}{\partial \rho_{\text{dy}}}\right|_{\rho_{\text{dy}}=0} - \eta \Lambda_t \underbrace{\left[ \frac{\sum_{\tau \in \mathcal{C}} \pi^{(t)}(\tau) \ln \pi^{(t)}(\tau)}{\sum_{\tau \in \mathcal{C}} \pi^{(t)}(\tau)} - \frac{\sum_{\tau \notin \mathcal{C}} \pi^{(t)}(\tau) \ln \pi^{(t)}(\tau)}{\sum_{\tau \notin \mathcal{C}} \pi^{(t)}(\tau)} \right]}_{\Delta \ell^{(t)}}.$$

From our definition of the discrepancy metric, we observe that $\text{Dis}(\pi^{(t)}) = -\Delta\ell^{(t)}$. Since $\Lambda_t$ is simply the value of $\text{Dis}(\pi^{(t)})$ treated as a constant, we substitute $\Lambda_t = -\Delta\ell^{(t)}$ into the recurrence:

$$\left.\frac{\partial C^{(t+1)}}{\partial \rho_{\text{dy}}}\right|_{\rho_{\text{dy}}=0} = \left.\frac{\partial C^{(t)}}{\partial \rho_{\text{dy}}}\right|_{\rho_{\text{dy}}=0} - \eta(-\Delta\ell^{(t)})(\Delta\ell^{(t)}) = \left.\frac{\partial C^{(t)}}{\partial \rho_{\text{dy}}}\right|_{\rho_{\text{dy}}=0} + \eta\left(\Delta\ell^{(t)}\right)^2.$$

Summing over the training trajectory from $t = 0$ to $T - 1$ (with $\frac{\partial C^{(0)}}{\partial \rho_{\text{dy}}} = 0$):

$$\left.\frac{\partial C^{(T)}}{\partial \rho_{\text{dy}}}\right|_{\rho_{\text{dy}}=0} = \eta \sum_{t=0}^{T-1} \left[ \frac{\sum_{\tau \in \mathcal{C}} \pi^{(t)}(\tau) \ln \pi^{(t)}(\tau)}{\sum_{\tau \in \mathcal{C}} \pi^{(t)}(\tau)} - \frac{\sum_{\tau \notin \mathcal{C}} \pi^{(t)}(\tau) \ln \pi^{(t)}(\tau)}{\sum_{\tau \notin \mathcal{C}} \pi^{(t)}(\tau)} \right]^2.$$

Since $\eta > 0$ and the squared term is non-negative (and strictly positive assuming the distributions on $\mathcal{C}$ and $\mathcal{C}^c$ are not identical in entropy), the sensitivity is strictly positive. $\qquad\square$

---

**Theorem 3.** *For any iteration $T > 0$, there exists a coefficient $\rho_{\text{dy}} > 0$ such that the policy trained via dynamic entropy scheduling yields a strictly higher value than the standard RL baseline, i.e., $V(\pi_{\text{dy}}^{(T)}) > V(\pi_{\text{van}}^{(T)})$.*

---

*Proof.* Invoking Lemma 11, we observe that the sensitivity of the correctness metric with respect to the dynamic entropy coefficient is strictly positive for any training step $T$:

$$\left.\frac{\partial C(\pi_{\text{dy}}^{(T)})}{\partial \rho_{\text{dy}}}\right|_{\rho_{\text{dy}}=0} > 0. \tag{20}$$

By the definition of the derivative, there exists a sufficiently small $\rho_{\text{dy}} > 0$ such that the objective value improves over the baseline ($\rho_{\text{dy}} = 0$). Consequently, we have $V(\pi_{\text{dy}}^{(T)}) > V(\pi_{\text{van}}^{(T)})$. $\qquad\square$

# C. Experimental Details for Section 6.3

## C.1. Experimental Setup

We use the dataset released by Pan et al. (2025), which contains 327,680 training samples and 1,024 test samples.[1] An example training prompt is shown below.

> **Countdown Task Example**
>
> A conversation between User and Assistant. The user asks a question, and the Assistant solves it. The assistant first thinks about the reasoning process in the mind and then provides the user with the answer.
> **User:** Using the numbers [5, 94, 9, 44], create an equation that equals 93. You can use basic arithmetic operations (+, -, *, /) and each number can only be used once. Show your work in `<think> </think>` tags. And return the final answer in `<answer> </answer>` tags, for example `<answer> (1 + 2) / 3 </answer>`.
> **Assistant:** Let me solve this step by step.
> `<think>`

Our implementation builds on the official repository of Pan et al. (2025)[2] and a fork adapted for A100 training.[3]

### C.1.1. TRAINING

We train with a global batch size of 128, with 5 rollouts per prompt, and use a mini-batch size of 64. The learning rate is $1 \times 10^{-6}$, and the KL penalty coefficient is set to $\beta_{\mathrm{KL}} = 1 \times 10^{-3}$. The reward is 1 for correct responses, 0.1 for incorrect yet properly formatted responses, and 0 for all others. The maximum response length is 1024 tokens. We perform RL fine-tuning of the `Qwen2.5-3B` model (Qwen Team, 2024) for 300 steps on 4 NVIDIA L40 GPUs.

| **Configuration for Countdown Task** | | | |
|---|---|---|---|
| **Parameter** | **Value** | **Parameter** | **Value** |
| Pretrained model | Qwen2.5-3B | Batch size | 128 |
| Generations/prompt | 5 | Mini-batch size | 64 |
| Max prompt length | 2,048 | Max response len | 1,024 |
| Learning rate | $1 \times 10^{-6}$ | Training steps | 300 |
| Entropy coeff | varies (see Sec 6.3) | Clip ratio | 0.2 |
| Rollout engine | `vllm` | Rollout temp | 1 |
| Validation temp | 1 | Validation top-k | 50 |
| Validation top-p | 0.7 | Device | NVIDIA L40 |

To evaluate performance across diverse problem settings, we vary the cardinality and magnitude of the input integers (number of operators, maximum operand magnitude) as well as the target value range. We benchmark the vanilla GRPO baseline against the GRPO variant with fixed entropy coefficients across these tasks.

---

[1] https://huggingface.co/datasets/Jiayi-Pan/Countdown-Tasks-3to4
[2] https://github.com/Jiayi-Pan/TinyZero
[3] https://github.com/JerryWu-code/TinyZero

# D. Experimental Details for Section 6.4

In this section, we provide the comprehensive experimental configurations and hyperparameter settings for the results presented in Section 6.4.

## D.1. Experimental Setup for Knights-and-Knaves Puzzles

We begin by detailing the training setup for the Knights-and-Knaves task.

### D.1.1. TASK DESCRIPTION

The Knights-and-Knaves puzzle is a constrained logical reasoning task derived from the ReasoningGym benchmark (Stojanovski et al., 2025). The environment consists of a group of inhabitants, each assigned one of two latent roles:

- **Knights**: Characters who strictly tell the truth.

- **Knaves**: Characters who strictly lie.

The model is presented with a set of inter-dependent statements made by these inhabitants. The objective is to deduce the correct identity (Knight or Knave) for every character based solely on the logical consistency of their statements. The difficulty of the task is determined by the number of inhabitants and the complexity of the logical chains required to reach a solution.

**Example Instance.** An illustrative example of a puzzle question and the corresponding correct response is provided below:

---

### Knights-and-Knaves Example

`<|im_start|>system`
You are a helpful assistant.`<|im_end|>`
`<|im_start|>user`
A very special island is inhabited only by sages and fools. Sages always tell the truth, and fools always lie. You meet 6 inhabitants: Owen, Liam, Evelyn, William, Sofia, and Emily.
As Owen put it, "William is a sage". Liam was heard saying, "William is a fool". Evelyn noted, "Sofia is a fool and Evelyn is a sage". "if Owen is a sage then Sofia is a sage," William mentioned. "if Liam is a sage then Evelyn is a sage," Sofia claimed. Emily told you that if Emily is a sage then Liam is a fool.
So who is a sage and who is a fool? (Format your answer like: "Owen is a sage/fool, Liam is a sage/fool, Evelyn is a sage/fool, William is a sage/fool, Sofia is a sage/fool, and Emily is a sage/fool") Let's think step by step and output the final answer (should be a sentence) within
`\boxed{}`. `<|im_end|>`
`<|im_start|>assistant`
Let's analyze the logical constraints step by step.

- **Analyze Owen and Liam:**

    - Owen says: "William is a sage."
    - Liam says: "William is a fool."
    - Since William cannot be both a sage and a fool, Owen and Liam must have opposite identities (one is a sage, one is a fool).

- **Analyze Emily's statement:** "If Emily is a sage then Liam is a fool."

    - If Emily is a *fool*: The statement must be false. The implication $P \rightarrow Q$ is only false if $P$ is true and $Q$ is false. This would mean "Emily is a sage" is true, which contradicts the assumption that she is a fool. Thus, Emily cannot be a fool.
    - Therefore, **Emily is a sage**. Since she is a sage, her statement is true. For "If sage then Liam is a fool" to be true (with the antecedent being true), the consequent must be true.
    - Therefore, **Liam is a fool**.

- **Back to Owen:**

    - We established Owen and Liam have opposite identities. Since Liam is a fool, **Owen is a sage**.

- **Check Owen's statement:**

    - Owen (Sage) says "William is a sage". This must be true.
    - Therefore, **William is a sage**.

- **Analyze William's statement:** "If Owen is a sage then Sofia is a sage."

    - William is a sage, so this statement is true.
    - We know Owen is a sage (Antecedent is True).
    - For the implication to be true, the Consequent must be true.
    - Therefore, **Sofia is a sage**.

---

- **Analyze Evelyn's statement:** "Sofia is a fool and Evelyn is a sage."
    - We know Sofia is a sage. Thus, the part "Sofia is a fool" is false.
    - Since one part of the conjunction ("and") is false, the whole statement is false.
    - Since Evelyn made a false statement, **Evelyn is a fool**.

**Summary of identities:**

- Owen: Sage

- Liam: Fool

- Evelyn: Fool

- William: Sage

- Sofia: Sage

- Emily: Sage

> Owen is a sage, Liam is a fool, Evelyn is a fool, William is a sage, Sofia is a sage, and Emily is a sage

`<|im_end|>`

### D.1.2. TRAINING DETAILS

We fine-tune Qwen2.5-1.5B-Instruct (Qwen Team, 2024) on a composite dataset spanning three difficulty levels (3–5). To ensure rigorous evaluation, we test the model on strictly disjoint test sets. The specific training configurations are detailed below:

| Configuration for Qwen2.5-1.5B-Instruct | | | |
|---|---|---|---|
| **Parameter** | **Value** | **Parameter** | **Value** |
| Pretrained Model | Qwen2.5-1.5B-Instruct | Training Set | KnK-3, KnK-4, KnK-5 |
| Prompts per batch | 32 | Generations/prompt | 8 |
| Grad update/RL step | 1 | Max prompt length | 1024 |
| Max response len | 2048 | Learning rate | $1 \times 10^{-6}$ |
| Clip ratio low | 0.2 | Clip ratio high | 0.25 |
| Training Steps | 500 | $\beta$ | 0.0 |
| Entropy coeff | 0.0 | $\gamma$ | 0.05 |
| Rollout temp | 1.0 | Validation top_p | 1.00 |
| Validation temp | 0.7 | Device | 4 x Nvidia-A6000 |

### D.2. Experimental Setup for Mathematical Reasoning

We detail the experimental configuration for the mathematical reasoning tasks. We fine-tune the Qwen2.5-Math-1.5B (Yang et al., 2024) and Qwen2.5-Math-7B (Yang et al., 2024) models using the DAPO17k dataset (Yu et al., 2025). Performance is evaluated across four established benchmarks: MATH500 (Lightman et al., 2024), OlympiadBench (He et al., 2024), AIME24 (math ai, 2024), and AIME25 (math ai, 2025). Representative response examples are provided in the boxes below.

---

**Qwen2.5-Math-1.5B**

`<|im_start|>system`
Please reason step by step, and put your final answer within `\boxed{}`. `<|im_end|>`
`<|im_start|>user`
Cities $A$ and $B$ are $45$ miles apart. Alicia lives in $A$ and Beth lives in $B$. Alicia bikes towards $B$ at 18 miles per hour. Leaving at the same time, Beth bikes toward $A$ at 12 miles per hour. How many miles from City $A$ will they be when they meet? Let's think step by step and output the final answer within `\boxed{}`. `<|im_end|>`
`<|im_start|>assistant`
To determine where Alicia and Beth will meet, we need to find out how long it will take for them to meet and then use that time to calculate the distance Alicia has traveled.

1. First, let's calculate their combined speed. Since Alicia bikes at 18 miles per hour and Beth bikes at 12 miles per hour, their combined speed is $18 + 12 = 30$ miles per hour.

2. Since the distance between the two cities is 45 miles, the time it takes for them to meet is the total distance divided by their combined speed. So, the time $t$ it takes for them to meet is

$$t = \frac{45}{30} = 1.5 \text{ hours.}$$

3. Now, we need to find out how far Alicia has traveled in 1.5 hours. Since Alicia bikes at 18 miles per hour, the distance she travels in 1.5 hours is

$$18 \times 1.5 = 27 \text{ miles.}$$

So, Alicia and Beth will meet 27 miles from City $A$. The final answer is

$$\boxed{27}$$

`<|im_end|>`

---

We present the detailed training setups for Qwen2.5-Math-1.5B (Yang et al., 2024) and Qwen2.5-Math-7B (Yang et al., 2024) as follows.

**Configuration for Qwen2.5-Math**

| Parameter | Value | Parameter | Value |
|---|---|---|---|
| Pretrained Model | Qwen2.5-Math | Training Set | DAPO17K |
| Prompts per batch | 32 | Generations/prompt | 8 |
| Grad update/RL step | 1 | Max prompt length | 1024 |
| Max response len | 2048 | Learning rate | $1 \times 10^{-6}$ |
| Clip ratio low | 0.2 | Clip ratio high | 0.25 |
| Training Steps | 500 | $\beta$ | 0.0 |
| Entropy coeff | 0.0 | $\gamma$ | 0.02 |
| Rollout temp | 0.7 | Validation top_p | 1.00 |
| Validation temp | 0.7 | Device | 4 x Nvidia-A6000 |

## D.3. Additional Experimental Detail

| Performance: Qwen2.5-Math-1.5B | | | | | | | |
|---|---|---|---|---|---|---|---|
| Dataset | Model | P@1 | P@2 | P@4 | P@8 | P@16 | P@32 | P@64 |
| OlympiadBench | GRPO | 32.58 | 40.36 | 47.33 | 53.63 | 59.42 | 64.51 | 69.19 |
| | **Ours** | 33.14 | 40.98 | 47.81 | 53.88 | 59.40 | 64.50 | 69.19 |
| | *Imp.* | **+0.56** | **+0.62** | **+0.48** | **+0.25** | -0.02 | -0.01 | 0.00 |
| MATH500 | GRPO | 68.49 | 76.90 | 83.06 | 87.60 | 91.01 | 93.38 | 94.80 |
| | **Ours** | 69.31 | 77.13 | 83.13 | 87.55 | 90.78 | 93.12 | 94.80 |
| | *Imp.* | **+0.82** | **+0.23** | **+0.07** | -0.05 | -0.23 | -0.26 | 0.00 |
| AIME24 | GRPO | 13.07 | 19.72 | 26.21 | 31.65 | 36.53 | 41.56 | 46.67 |
| | **Ours** | 13.33 | 19.89 | 26.10 | 31.59 | 37.45 | 44.58 | 53.33 |
| | *Imp.* | **+0.26** | **+0.17** | -0.11 | -0.06 | **+0.92** | **+3.02** | **+6.66** |
| AIME25 | GRPO | 6.14 | 10.35 | 15.50 | 20.36 | 26.26 | 33.65 | 43.33 |
| | **Ours** | 6.78 | 11.33 | 17.13 | 23.43 | 30.13 | 38.55 | 50.00 |
| | *Imp.* | **+0.64** | **+0.98** | **+1.63** | **+3.07** | **+3.87** | **+4.90** | **+6.67** |

| Performance: Qwen2.5-Math-7B | | | | | | | |
|---|---|---|---|---|---|---|---|
| Dataset | Model | P@1 | P@2 | P@4 | P@8 | P@16 | P@32 | P@64 |
| OlympiadBench | GRPO | 41.33 | 48.63 | 55.21 | 60.75 | 65.29 | 69.16 | 72.81 |
| | **Ours** | 42.19 | 49.39 | 55.82 | 61.34 | 66.00 | 70.15 | 74.00 |
| | *Imp.* | **+0.86** | **+0.76** | **+0.61** | **+0.59** | **+0.71** | **+0.99** | **+1.19** |
| MATH500 | GRPO | 78.52 | 84.12 | 87.82 | 90.51 | 92.71 | 94.41 | 95.40 |
| | **Ours** | 79.18 | 84.37 | 87.96 | 90.67 | 92.88 | 94.39 | 95.40 |
| | *Imp.* | **+0.66** | **+0.25** | **+0.14** | **+0.16** | **+0.17** | -0.02 | 0.00 |
| AIME24 | GRPO | 24.95 | 34.62 | 43.19 | 49.16 | 53.15 | 56.67 | 60.00 |
| | **Ours** | 27.44 | 35.90 | 42.92 | 49.29 | 55.03 | 59.96 | 63.33 |
| | *Imp.* | **+2.49** | **+1.28** | -0.27 | **+0.13** | **+1.88** | **+3.29** | **+3.33** |
| AIME25 | GRPO | 12.71 | 18.31 | 24.37 | 30.22 | 35.37 | 40.83 | 46.67 |
| | **Ours** | 13.54 | 18.48 | 23.30 | 27.86 | 33.16 | 40.08 | 46.67 |
| | *Imp.* | **+0.83** | **+0.17** | -1.07 | -2.36 | -2.21 | -0.75 | 0.00 |

