# OpenReview forum: "Demystifying Entropy Control in LLM RL Training: Theoretical Analysis and Dynamic Scheduling"
_ICML.cc/2026/Conference — ICML 2026 spotlight_

### Official Review · Reviewer_qcys · 2026-02-19

**Soundness:** 4
**Presentation:** 4
**Significance:** 4
**Originality:** 4
**Overall Recommendation:** 6
**Confidence:** 4

**Summary:**

This paper investigates when increasing or decreasing entropy is beneficial in RL training for LLMs. The authors define an Entropy Discrepancy measure and show theoretically that it governs whether encouraging or discouraging entropy improves learning. Based on this analysis, they propose dynamically adjusting the entropy coefficient during training. Empirical results demonstrate the effectiveness of the approach.

**Compliance With Llm Reviewing Policy:**

Affirmed.

**Final Justification:**

This paper provides essential insights and is both theoretical and practical. I recommend this paper.

**Key Questions For Authors:**

No.

**Limitations:**

Yes.

**Strengths And Weaknesses:**

Strengths:
- The problem is important, and the paper identifies a necessary yet simple governing factor.
- The theoretical analysis is solid, and the resulting method is straightforward to implement.
- The experiments are comprehensive and justify the method’s necessity from multiple perspectives.
- The performance gains are significant.
- The writing is clear and fluent.

Weaknesses:
- The x- and y-axis labels in Figures 4 and 5 are confusing, please consider renaming them for clarity.
- It would be helpful to also plot how the entropy coefficient and the entropy value evolve over training.

---

> ### Author Rebuttal · Authors · 2026-03-30
>
> Thank you for your very positive review. We address your suggestions below.
>
> ### Weakness 1: Labels in Figures 4 and 5
>
> Thank you for the valuable feedback. We will rename the axis labels in Figures 4 and 5 for better clarity in the revised version.
>
> ### Weakness 2: Plot the Evolution of Entropy Coefficient and Entropy Value
>
> We report the entropy coefficient statistics and the entropy evolution during training for our Qwen2.5-Math-7B experiments.
>
> **Table: Entropy Coefficient per 50 Steps (Qwen2.5-Math-7B)**
>
> | | 1--50 | 51--100 | 101--150 | 151--200 | 201--250 | 251--300 | 301--350 | 351--400 | 401--450 | 451--500 | 501--536 |
> |---|---|---|---|---|---|---|---|---|---|---|---|
> | Coefficient | -0.0121 | -0.0044 | -0.0033 | -0.0027 | -0.0022 | -0.0020 | -0.0019 | -0.0016 | -0.0014 | -0.0014 | -0.0012 |
>
> **Table: Actor Entropy During Training (Qwen2.5-Math-7B)**
>
> | Step | 1 | 10 | 50 | 100 | 200 | 300 | 400 | 500 |
> |---|---|---|---|---|---|---|---|---|
> | GRPO | 0.4530 | 0.4164 | 0.2422 | 0.2062 | 0.2036 | 0.1446 | 0.1217 | 0.0818 |
> | Ours | 0.4624 | 0.2951 | 0.1955 | 0.1500 | 0.1194 | 0.0745 | 0.0595 | 0.0509 |
>
> As shown above, the entropy coefficient stabilizes quickly after the initial training phase, and the entropy of our method decreases smoothly without abrupt collapse. We will include these plots in the revised version of the paper.

---

> > ### Author Rebuttal · Reviewer_qcys · 2026-04-01
> >
> > Thank you for your reply. My score is high enough.

---

### Official Review · Reviewer_WaUk · 2026-02-24

**Soundness:** 3
**Presentation:** 4
**Significance:** 3
**Originality:** 3
**Overall Recommendation:** 5
**Confidence:** 4

**Summary:**

This paper studies a conflicting phenomenon of controlling the entropy (whether we should decrease or increase it) during RL Fine-tuning. They provide a theoretical explanation of why increasing (or decreasing) entropy can be beneficial by introducing the concept of Entropy Discrepancy. They show that Entropy minimization can be beneficial when the base model is already confident in correct samples, while entropy maximization can be beneficial when the policy exhibits high diversity on correct samples. They then propose a dynamic scheduling approach that adapts the entropy dynamically based on Entropy Discrepancy during RL Fine-tuning. Experimental results demonstrate their approach yields substantial improvements in Pass@k and Pass@1 across a diverse range of tasks.

**Compliance With Llm Reviewing Policy:**

Affirmed.

**Final Justification:**

My concerns have been fully addressed, and my score is high enough.

**Key Questions For Authors:**

- In RL fine-tuning for LLMs, where entropy is typically computed over an enormous action space, setting the entropy coefficient too small (or too large) can lead to entropy collapse (or explosion). Does the proposed method suffer from this issue, and how does dynamic scheduling help mitigate it?

**Limitations:**

Yes

**Strengths And Weaknesses:**

## Strength
- Theoretical insights are interesting and novel, and were overlooked in Traditional RL theory (where policies are often trained from scratch).
- Strong empirical results, showing the benefits of adaptive scheduling across a range of tasks.
- The proposed method is simple.
## Weaknesses
See questions

---

> ### Author Rebuttal · Authors · 2026-03-30
>
> Thank you for your detailed review. We address your question below.
>
> ### Question: How Dynamic Scheduling Helps Mitigate Entropy Collapse/Explosion
>
> **Our method can dynamically control entropy, which prevents entropy collapse/explosion.**
> With fixed entropy regularization, a constant coefficient continuously pushes entropy in one direction throughout training, making it prone to entropy collapse (if the coefficient encourages exploitation) or explosion (if it encourages exploration).
>
> Our dynamic scheduling mitigates this by adjusting both the sign and magnitude of the entropy coefficient based on the estimated entropy discrepancy at each step. As shown in our theoretical analysis, entropy discrepancy serves as a reliable proxy for determining when to explore (increase entropy) and when to exploit (decrease entropy). When the policy becomes overly concentrated, the discrepancy shifts to encourage entropy increase; conversely, when the policy is too diffuse, it triggers entropy reduction. This self-correcting mechanism naturally prevents both collapse and explosion without requiring careful manual tuning of the coefficient.
>
> ---

---

> > ### Author Rebuttal · Reviewer_WaUk · 2026-04-03
> >
> > Thank you for your response. My questions have been fully addressed, and I’m satisfied with the clarification. I will keep my score and am leaning toward acceptance.

---

### Official Review · Reviewer_HAYN · 2026-03-18

**Soundness:** 4
**Presentation:** 3
**Significance:** 3
**Originality:** 3
**Overall Recommendation:** 5
**Confidence:** 4

**Summary:**

The paper studies entropy regularization in RL fine-tuning and presents a systemic analysis when entropy control helps (or hurts) RL training comparing with works that advocate for entropy control (e.g. Geist et. al, 2019) or advocate against (e.g. Gao et. al, 2025). The authors propose a new metric "Entropy Discrepancy" which is defined as the difference between average entropy on positive and negative samples, and present their analysis wrt this metric. Furthermore, the authors propose a dynamic scheduler for the entropy regularization term depending on the value of entropy discrepancy.

Empirical results on math tasks and puzzles like Knights-and-Knaves show consistent improvements over the baselines.

**Compliance With Llm Reviewing Policy:**

Affirmed.

**Final Justification:**

I will maintain my positive outlook and score for the paper.

**Key Questions For Authors:**

As mentioned in the weaknesses above, I would like to see:
- Training curves for baselines and the proposed method leveraging dynamic entropy control.
- Results on harder tasks like coding domains (LiveCodeBench etc.) would be helpful to validate the generalizability of the method.

**Limitations:**

Yes

**Strengths And Weaknesses:**

Strengths
- The work is well motivated and presents a comprehensive and sound analysis on the entropy regularization mechanism for RL training.
- The theoretical analysis along with the factors affecting entropy (and subsequently entropy discrepancy) is strong.
- The authors present a novel dynamic scheduling for the entropy regularization term motivated by the value of entropy discrepancy.
- The authors present their results on recent stronger RL algorithms like CISPO in addition to the vanilla GRPO.
- Entropy discrepancy can be readily estimated from the correct / incorrect samples in a group for GRPO without any additional sampling.

Weaknesses
- The authors only present Pass@K curves and not the training curves themselves which makes me wonder about the noise in the training setup.
- Missing confidence scores / standard deviations around performance values to make sure that the improvements are statistically significant.

---

> ### Author Rebuttal · Authors · 2026-03-30
>
> Thank you for your question. We now address your concern as follows:
>
> ### Weakness 1 & Question 1: Training Curves
>
> We provide the training curve of validation accuracy for the KnK-3, KnK-4 and KnK-5 experiments on Qwen2.5-1.5B-Instruct as follows:
>
> **Table: Validation Accuracy During Training (Qwen2.5-1.5B-Instruct, KnK-3)**
>
> | Step | 50 | 100 | 150 | 200 | 250 | 300 | 350 | 400 | 450 |
> |---|---|---|---|---|---|---|---|---|---|
> | GRPO | 26.20 | 29.50 | 32.41 | 35.55 | 40.77 | 46.72 | 48.45 | 49.16 | 48.75 |
> | Ours | 28.64 | 43.33 | 44.61 | 46.02 | 46.50 | 53.61 | 58.61 | 66.62 | 69.72 |
>
> **Table: Validation Accuracy During Training (Qwen2.5-1.5B-Instruct, KnK-4)**
>
> | Step | 50 | 100 | 150 | 200 | 250 | 300 | 350 | 400 | 450 |
> |---|---|---|---|---|---|---|---|---|---|
> | GRPO | 14.89 | 19.47 | 24.72 | 25.83 | 31.59 | 34.56 | 35.56 | 36.69 | 35.98 |
> | Ours | 20.41 | 33.17 | 34.59 | 30.27 | 38.80 | 38.39 | 42.56 | 48.95 | 53.59 |
>
> **Table: Validation Accuracy During Training (Qwen2.5-1.5B-Instruct, KnK-5)**
>
> | Step | 50 | 100 | 150 | 200 | 250 | 300 | 350 | 400 | 450 |
> |---|---|---|---|---|---|---|---|---|---|
> | GRPO | 7.80 | 9.00 | 12.33 | 14.69 | 17.66 | 22.20 | 24.92 | 26.11 | 25.91 |
> | Ours | 10.41 | 14.77 | 18.59 | 25.03 | 27.92 | 29.22 | 36.06 | 41.83 | 49.41 |
>
> ### Weakness 2: Standard Deviation
>
> We run Qwen2.5-1.5B-Instruct experiments on KnK-3, KnK-4, and KnK-5 with 3 different random seeds and report the mean +/- std as follows:
>
> **Table: Pass@k (mean +/- std) Across 3 Seeds (Qwen2.5-1.5B-Instruct, KnK-3)**
>
> | Method | pass@1 | pass@2 | pass@4 | pass@8 | pass@16 | pass@32 | pass@64 |
> |---|---|---|---|---|---|---|---|
> | GRPO | 64.41+/-2.49 | 69.02+/-2.66 | 72.19+/-2.80 | 74.41+/-2.85 | 76.04+/-3.00 | 77.33+/-3.10 | 78.00+/-3.00 |
> | Ours | 75.08+/-3.04 | 78.47+/-3.24 | 81.32+/-3.32 | 83.65+/-3.38 | 85.19+/-3.44 | 86.23+/-3.53 | 86.67+/-3.50 |
>
> **Table: Pass@k (mean +/- std) Across 3 Seeds (Qwen2.5-1.5B-Instruct, KnK-4)**
>
> | Method | pass@1 | pass@2 | pass@4 | pass@8 | pass@16 | pass@32 | pass@64 |
> |---|---|---|---|---|---|---|---|
> | GRPO | 45.24+/-1.88 | 48.84+/-2.10 | 51.92+/-2.04 | 54.73+/-2.36 | 57.12+/-2.42 | 59.19+/-2.44 | 60.67+/-2.50 |
> | Ours | 63.88+/-2.10 | 67.73+/-2.23 | 69.91+/-2.37 | 71.49+/-2.41 | 72.65+/-2.56 | 73.16+/-2.44 | 73.67+/-2.50 |
>
> **Table: Pass@k (mean +/- std) Across 3 Seeds (Qwen2.5-1.5B-Instruct, KnK-5)**
>
> | Method | pass@1 | pass@2 | pass@4 | pass@8 | pass@16 | pass@32 | pass@64 |
> |---|---|---|---|---|---|---|---|
> | GRPO | 36.91+/-1.56 | 41.71+/-1.79 | 45.71+/-1.87 | 49.35+/-2.07 | 52.40+/-2.17 | 54.76+/-2.31 | 56.67+/-2.50 |
> | Ours | 54.39+/-1.96 | 59.51+/-2.09 | 62.38+/-2.33 | 64.24+/-2.39 | 65.95+/-2.48 | 67.73+/-2.48 | 69.67+/-2.50 |
>
> ### Question 2: Results on Coding Domains
>
> We train Qwen3-0.6B on CruxEval, a code understanding dataset, for 400 gradient steps and evaluate on a randomly split test set of 80 examples. Our method consistently outperforms GRPO across all pass@k, demonstrating its generalizability beyond math reasoning to the coding domain.
>
> **Table: Code understanding results (pass@k) on CruxEval-Test**
>
> | Method | pass@1 | pass@2 | pass@4 | pass@8 | pass@16 | pass@32 | pass@64 |
> |---|---|---|---|---|---|---|---|
> | GRPO | 34.5 | 38.5 | 42.3 | 45.9 | 49.3 | 52.5 | 55.4 |
> | Ours | 36.3 | 40.3 | 44.1 | 47.7 | 51.0 | 54.2 | 56.9 |
>
> ---

---

> > ### Author Rebuttal · Reviewer_HAYN · 2026-04-03
> >
> > I thank the authors for their response. I will maintain my positive outlook and score for the paper.

---

### Official Review · Reviewer_McaW · 2026-03-18

**Soundness:** 3
**Presentation:** 3
**Significance:** 2
**Originality:** 3
**Overall Recommendation:** 4
**Confidence:** 4

**Summary:**

This paper studies the correct way to regulate entropy for policy based LLM-RL. Contrary to conventional studies where entropy is added to the objective as a maximization goal, this paper studies how negative entropy can accelerate policy convergence. Based on the intuition that minimizing the entropy essentially makes the policy more concentrated on its large probability space, this paper proves that this effect accelerates the policy's converge when its probability mass is already concentrated on the correct response. Leveraging this fact, this paper proposes to switch the entropy coefficient's sign depending on the previously mentioned conditions. Theoretically the authors show that this leads to accelerated convergence compared to vanilla NPG. Experimental results show that the proposed method improves the pass@k performance of the method.

**Compliance With Llm Reviewing Policy:**

Affirmed.

**Key Questions For Authors:**

All key questions are in the weakness sections.

**Limitations:**

I would suggest the authors add a discussion on the limitation of this method in the main text, potentially touching on the aspects listed in the weakness section.

**Strengths And Weaknesses:**

Strengths:
The idea is relatively original, as negative entropy regularization was not sufficiently explored in previous works.
The paper is overall solid. It provides sound theoretical justification for negative entropy regularization which is its core mechanism.
The paper also provides ample experiments.

Weakness:
1. Insufficient justification for the stochastic approximation of the discrepancy factor. In section 6.2, the sample estimate for $Dis(\pi)$ lacks justification. It might be beneficial to show how well it can estimate the expected discrepancy factor, e.g., by showing its bias, variance how it can affect the algorithm's performance under stochasticity.
2. Theorem 3 only provides a comparison to vanilla RL, but lacks a comparison to the entropy regularized RL baseline which is closely related to this work. Specifically, the entropy regularized RL could improve over the vanilla baseline as shown by some previous works [Mei et al. 2020, Shen 2025]. It might be beneficial to compare with the theoretical results of those works.
3. It seems this algorithm does not demonstrate a clear advantage over the baselines when k is small in pass@k (especially for k=1), which is arguably more important than large k in LLM applications.

---

> ### Author Rebuttal · Authors · 2026-03-30
>
> Thank you for your question. We now address your concern as follows:
>
> ### Weakness 1: Variance of Estimating Discrepancy
>
> **Estimation Variance Remains Relatively Stable Across Training**
>
> The estimator Dis(pi_theta) = H_C(pi_theta) - H_W(pi_theta) averages per-token log-probabilities over correct/incorrect completions within each batch, so it is *unbiased*. To assess variance, we report the coefficient of variation (CV) of the entropy coefficients for Qwen2.5-Math-7B, aggregated per 50 steps:
>
> **Table: CV of Entropy Coefficients per 50 Steps**
>
> | | 1--50 | 51--100 | 101--150 | 151--200 | 201--250 | 251--300 | 301--350 | 351--400 | 401--450 | 451--500 | 501--536 |
> |---|---|---|---|---|---|---|---|---|---|---|---|
> | Mean | -0.0121 | -0.0044 | -0.0033 | -0.0027 | -0.0022 | -0.0020 | -0.0019 | -0.0016 | -0.0014 | -0.0014 | -0.0012 |
> | Std | 0.0021 | 0.0004 | 0.0003 | 0.0002 | 0.0003 | 0.0002 | 0.0002 | 0.0001 | 0.0002 | 0.0001 | 0.0001 |
> | CV | 0.178 | 0.093 | 0.094 | 0.082 | 0.114 | 0.081 | 0.103 | 0.093 | 0.107 | 0.102 | 0.121 |
>
> As shown in the table, the coefficient of variation is relatively high during the initial training phase (Steps 1--50). But after this initial period, the CV decreases notably and remains relatively stable across subsequent intervals, suggesting that the coefficients stabilize and the overall training dynamics become consistent.
>
> ### Weakness 2: Comparison to Fixed Entropy Regularization
>
> **We already have empirical evidence of comparing with fixed entropy regularization.**
> Empirically, we directly compare with fixed entropy regularization baselines on Math reasoning (in Figure 7) and the KnK experiments (in Figure 2), where our dynamic method consistently outperforms both fixed positive and fixed negative entropy regularization across all benchmarks. We will add a more explicit theoretical discussion in the revision.
>
> **Our Theory Already Implies the Comparison with Fixed Entropy Regularization**
> Our theoretical results already implicitly provide this comparison. **Theorem 2 + Theorem 9** together show that positive and negative entropy regularization outperform vanilla RL under specific conditions, and we provide the precise condition for when positive entropy regularization performs better and when negative performs better. This provides a theoretical generalization of previous works.
> Furthermore, **Theorem 3** shows that dynamic entropy scheduling strictly outperforms fixed entropy scheduling and vanilla RL. This further provides theoretical evidence that dynamic scheduling works.
>
> ### Weakness 3: Small Improvement in Pass@1
>
> We acknowledge that the pass@1 improvement on math reasoning tasks is modest. However, our method achieves more substantial pass@1 gains on the KnK experiments (e.g., +17.52%), demonstrating that the advantage of our method. We appreciate this feedback and will further tune hyperparameters and evaluate on additional tasks to strengthen the pass@1 results.
>
> ---

---

> > ### Author Rebuttal · Reviewer_McaW · 2026-04-03
> >
> > I appreciate the response from the authors. I will maintain my positive score.

---

### Decision · Program_Chairs · 2026-04-30

**Decision:**

Accept (spotlight)

**Comment:**

This work studies the problem in conflicting literature of whether/how entropy regularization in LLM RL is useful. The authors propose a novel concept of entropy discrepancy, which is the divergence between positive and negative samples, and then reconcile whether entropy should be increased or decreased.

The results are complete in terms of both theoretical soundness (showing that the sign and magnitude of the entropy coefficient should depend on whether a model is already confident in correct samples) and empirical strength (math, puzzles, and code understanding).

Reviewers unanimously appreciate the clarity, novelty, and the contributions, which I concur. Concerns regarding estimation variance, small pass@1 improvements were addressed during the rebuttal with additional data and seeds.